# Provably Safe Reinforcement Learning: Conceptual Analysis, Survey, and Benchmarking

**Hanna Krasowski**[*]                    *hanna.krasowski@tum.de*

**Jakob Thumm**[*]                        *jakob.thumm@tum.de*

**Marlon Müller**                         *marlon.mueller@tum.de*

**Lukas Schäfer**                         *lukas.schaefer@tum.de*

**Xiao Wang**                             *xiao.wang@tum.de*

**Matthias Althoff**                      *althoff@tum.de*

*School of Computation, Information and Technology*
*Technical University of Munich*

**Reviewed on OpenReview:** *https://openreview.net/forum?id=mcN0ezbnzO*

## Abstract

Ensuring the safety of reinforcement learning (RL) algorithms is crucial to unlock their potential for many real-world tasks. However, vanilla RL and most safe RL approaches do not guarantee safety. In recent years, several methods have been proposed to provide hard safety guarantees for RL, which is essential for applications where unsafe actions could have disastrous consequences. Nevertheless, there is no comprehensive comparison of these provably safe RL methods. Therefore, we introduce a categorization of existing provably safe RL methods, present the conceptual foundations for both continuous and discrete action spaces, and empirically benchmark existing methods. We categorize the methods based on how they adapt the action: action replacement, action projection, and action masking. Our experiments on an inverted pendulum and a quadrotor stabilization task indicate that action replacement is the best-performing approach for these applications despite its comparatively simple realization. Furthermore, adding a reward penalty, every time the safety verification is engaged, improved training performance in our experiments. Finally, we provide practical guidance on selecting provably safe RL approaches depending on the safety specification, RL algorithm, and type of action space.

## 1 Introduction

Reinforcement learning (RL) contributes to many recent advancements in challenging research fields such as robotics (El-Shamouty et al., 2020; Zhao et al., 2020), autonomous systems (Kiran et al., 2022; Ye et al., 2021), and games (Mnih et al., 2013; Silver et al., 2017). A vanilla RL agent typically explores randomly and executes undesired actions multiple times to learn how to achieve the highest possible reward. However, safety is important for many applications. Therefore, safe RL emerged where the learning process is adapted such that the agent considers safety aspects next to performance during training and/or operation (García & Fernández, 2015). There are different degrees of how safety is considered for safe RL approaches. First, some approaches only incorporate safety aspects without formal guarantees. Here, the agent chooses safe actions with higher probability, e.g., by adding a reward component that indicates risk or adapting the exploration such that the agent follows a safe heuristic. Second, some approaches provide probabilistic safety guarantees, e.g., by using probabilistic models for safe actions (Könighofer et al., 2021). Still, hard safety guarantees for RL agents are necessary whenever failures are disastrous and need to be avoided at all costs during training and deployment. Such safety-critical applications include autonomous driving, human-robot collaboration,

---

[*]Equal contribution.

or energy grids. We refer to this third subcategory of safe RL methods providing hard safety guarantees for both training and operation as *provably safe RL*.

We provide for the first time a consistent conceptual framework for *provably safe RL* in both continuous and discrete action spaces, a comprehensive literature survey, and a comparison between provably safe RL approaches on two widely used control benchmarks. The characteristic difference between provably safe RL approaches is how they adapt the actions of the agent. Therefore, we propose classifying them into three categories: *action replacement*, *action projection*, and *action masking*. Our proposed categorization of provably safe RL provides a concise presentation of the research field, supports researchers implementing provably safe RL through clear terminology and a comprehensive literature review, and outlines ideas for future research within the three categories. Furthermore, we evaluate the methods experimentally. Our three main findings of our experiments were that all provably safe RL methods were indeed safe, that action replacement performed best on average over five tested RL algorithms, and that adding a penalty to the reward when using the safety function further improved performance.

Our contributions are fourfold. First, we introduce a comprehensive classification of provably safe RL methods and their formal description. This categorization allows us to compare and benchmark the effects of choosing a specific type of action modification on the ability of agents to learn. Second, we propose the first formulation of action masking for continuous action spaces. Third, we provide a structured and comprehensive survey of previous provably safe RL works and assign them to the three categories. Finally, we are the first to evaluate the performance of all three provably safe RL methods on two common control benchmarks. This comparison provides insights into the strengths and weaknesses of the different provably safe RL approaches and allows us to provide advice on selecting the best-suited provably safe RL approach for a specific problem independent of the safety verification method used.

The remainder of this paper is structured as follows. First, we briefly review the historical development of safe RL and provably safe RL in Section 1.1. We describe preliminary concepts in Section 2 and introduce our proposed categorization. Then, we show how the related provably safe RL literature fits our categorization in Section 3. Section 4 compares the different provably safe RL categories experimentally on a two-dimensional (2D) quadrotor stabilization task. Section 5 discusses the results of our experimental evaluation and the practical considerations following them. Finally, we conclude this work in Section 6.

## 1.1 Evolution towards provably safe RL

The notion of risk and safety in RL is discussed at least since the 1990s (Heger, 1994). The reasons for combining safety and RL were to focus the learning on relevant or safe regions and improve the convergence speed. Thus, the field of safe RL started developing, and in 2015, García & Fernández (2015) were the first to cluster safe RL. They provide two high-level categories: approaches that modify the optimization criterion and approaches that modify the exploration. Since 2015, significant advances in model-free RL and the increased applicability of deep RL changed the research focus of safe RL. Notably, the higher efficiency of model-free deep RL made its real-world application tangible and amplified the need for formal guarantees in safe RL. This is also apparent from the survey by Brunke et al. (2022), who investigated recent developments at the intersection of control and learning for safe robotics. As a goal of the broader field of safe learning for control, they identify methods with as little as possible system knowledge while ensuring formal safety guarantees (Brunke et al., 2022, Fig. 4). Among existing safe RL approaches, provably safe RL research is a growing field located at this frontier as it provides hard safety guarantees during both learning and deployment. While a few papers mentioned in Brunke et al. (2022) are part of provably safe RL, it is not a focus of their work. In the following paragraphs, we use the common classification by safety specification type, which can be *soft constraints*, *probabilistic guarantees*, and *hard guarantees*, to locate provably safe RL in the field of safe RL.

**Soft constraints** Approaches with soft constraints consider safety directly in their optimization objective so that the agent can explore all actions and states regardless of safety. Thus, these methods can be unsafe during training, especially initially, but usually converge to a safer policy without formal safety guarantees after sufficiently many training steps. The simplest way to inform an RL agent about safety constraints is through its reward function. Despite its elegance, the reward function approach has many potential pitfalls.

First, the reward function might be ill-defined, either from manual tuning or when learned from human input. When manually defined, the reward function might overlook certain features or fine details, leading to a hackable reward (Skalse et al., 2022) from which the agent learns an unsafe behavior. Learning the reward function from human feedback (Christiano et al., 2017) is also error-prone because communicating safety constraints alongside performance metrics is hard for sparse, nonlinear, conditional, or seldom occurring constraints. Second, even if the reward function is defined correctly, the trained policy is not guaranteed to be safe, e.g., it was shown by Packer et al. (2018) that RL agents struggle with out-of-distribution states during deployment. Third, the agent might learn to perform actions safely but ignore the task objective due to goal misgeneralization (Langosco et al., 2022). Still, the majority of safe RL research considers safety aspects as soft constraints, so we provide a short overview of soft constraint methods in the following paragraph.

To reduce the burden of manual reward specification, a recent line of work formalizes the task and its safety specifications as a temporal logic formula. Then, the temporal logic formula is transformed into the RL reward either by directly using the robustness measure associated with the formula as the reward (Aksaray et al., 2016; Li et al., 2017; Varnai & Dimarogonas, 2020) or by transforming the temporal logic formula into an automaton that generates the reward (Camacho et al., 2019; Hahn et al., 2019; Cai et al., 2021; Hasanbeig et al., 2022; Alur et al., 2023). For some algorithms, it can even be ensured that the policy converges to the optimal policy, which maximally satisfies the temporal logic specification (Alur et al., 2022; Yang et al., 2022). Another way to inform an RL agent about safety than through the reward is by formulating a constrained optimization problem. Many recent advances have been made in constrained RL (Altman, 1998; Achiam et al., 2017; Stooke et al., 2020), for which the policy aims to maximize the reward while satisfying user-defined specifications. The specifications can be formulated as constraint functions (Chow et al., 2018; Stooke et al., 2020; Yang et al., 2020; Marvi & Kiumarsi, 2021) or as temporal logic formulas (De Giacomo et al., 2021; Hasanbeig et al., 2019a;b; 2020). The main advantage of soft constraint methods over probabilistic or hard constraint methods is that no explicit model of the agent dynamics or the environment is required as the agent learns the safety aspects through experience. Thus, such safe RL methods have a high potential in non-critical settings, where unsafe actions do not cause major damage.

**Probabilistic guarantees**  Probabilistically safe RL approaches rely on probabilistic models or synthesize a model from sampled data. Here, the action and state space can be restricted based on probabilities. Nonetheless, unsafe actions are sometimes not detected and might occur occasionally. Several works (Turchetta et al., 2016; Berkenkamp et al., 2017; Mannucci et al., 2018) try to determine the maximal set of safe states by starting from an often user-defined conservative set and extending it with the gathered learning experience. Other methods (Könighofer et al., 2021; Thananjeyan et al., 2021; Dalal et al., 2018; Zanon & Gros, 2021; Yang et al., 2021; Gillula & Tomlin, 2013) are based on formulating probabilistic models that identify the probability of safety for an action. In general, approaches that rely on probabilistic methods are especially applicable if one cannot bound measurement errors, modeling errors, and disturbances by sets.

**Hard guarantees**  Provably safe RL features hard safety guarantees, which are fulfilled by integrating prior system knowledge into the learning process. Here, the agent only explores safe actions and only reaches states fulfilling the safety specifications. Provably safe RL already fulfills the given safety specifications during the learning process, which is essential when training or fine-tuning agents on safety-critical tasks in the physical world. Thus, we exclude approaches that only verify learned policies (Bastani et al., 2018; Schmidt et al., 2021) from our survey. We focus on model-free RL algorithms that do not explicitly learn or use a model of the system dynamics to optimize the policy. Generally, deploying learned controllers in the physical world became increasingly realistic in recent years, and thus, the need for provably safe RL grew, and more provably safe RL approaches were developed. With this work, we aim to structure and provide practical insights into this growing field.

## 2  Conceptual analysis

We introduce three provably safe RL classes by providing their formal description in one comprehensive conceptual framework. This framework clarifies the differences between the three classes and eases the following literature review and benchmarking.

**Markov decision process**  The RL agent learns on a Markov decision process (MDP) that is described by the tuple $(\mathbb{S}, \mathbb{A}, T, r, \gamma)$. Hereby, we assume that the set of states $\mathbb{S}$ is fully observable with bounded precision. Partially observable MDPs can be handled using methods like particle filtering (Sunberg & Kochenderfer, 2018) and are not further discussed in this work. The action space $\mathbb{A}$ and state space $\mathbb{S}$ can be continuous or discrete. $T(\boldsymbol{s}, \boldsymbol{a}, \boldsymbol{s}')$ is the transition function, which in the discrete case returns the probability that the transition from state $\boldsymbol{s}$ to state $\boldsymbol{s}'$ occurs by taking action $\boldsymbol{a}$. In the continuous case, $T(\boldsymbol{s}, \boldsymbol{a}, \boldsymbol{s}')$ denotes the probability density function of the transition. We assume that the transition function is stationary over time. For each transition, the agent receives a reward $r : \mathbb{S} \times \mathbb{A} \to \mathbb{R}$ from the environment. The discount factor $0 < \gamma < 1$ weights the relevance of future rewards. The policy or value function that the action learns can be optimized for an infinite or finite episode horizon $p$.

**Safety of a system**  For provably safe RL, it is required that the safety of states and actions is verifiable. Otherwise, no formal claims about the safety of a system can be made. Thus, we first introduce the set of safe states $\mathbb{S}_{\boldsymbol{s}} \subseteq \mathbb{S}$ containing all states for which all safety specifications are fulfilled[1]. For verifying the safety of actions, we use a safety function $\varphi : \mathbb{S} \times \mathbb{A} \to \{0, 1\}$

$$\varphi(\boldsymbol{s}, \boldsymbol{a}) = \begin{cases} 1, & \text{if } (\boldsymbol{s}, \boldsymbol{a}) \text{ is verified safe} \\ 0, & \text{otherwise.} \end{cases} \tag{1}$$

Conceptually, this is mainly done by over-approximating the set of states that are reachable by taking action $\boldsymbol{a}$ in state $\boldsymbol{s}$, and then validating if the reachable set of states is a subset of $\mathbb{S}_{\boldsymbol{s}}$ and if all these reachable states are verified safe until the episode horizon $p$. In other words, for each of these reachable states, there exists at least one action that keeps the system within $\mathbb{S}_{\boldsymbol{s}}$ until episode termination. To formalize this concept, we define the set of provably safe actions $\mathbb{A}_{\varphi}(\boldsymbol{s}) = \{\boldsymbol{a} | \varphi(\boldsymbol{s}, \boldsymbol{a}) = 1\}$ for a given state $\boldsymbol{s}$. The set of provably safe actions $\mathbb{A}_{\varphi}(\boldsymbol{s})$ is a subset of all safe actions[2] $\mathbb{A}_{\boldsymbol{s}}(\boldsymbol{s})$, i.e., $\mathbb{A}_{\varphi}(\boldsymbol{s}) \subseteq \mathbb{A}_{\boldsymbol{s}}(\boldsymbol{s}) \subseteq \mathbb{A}$. The safe action set $\mathbb{A}_{\boldsymbol{s}}(\boldsymbol{s})$ includes all safe actions, while the provably safe action set $\mathbb{A}_{\varphi}(\boldsymbol{s})$ only includes actions that are verified as safe by the safety function $\varphi(\boldsymbol{s}, \boldsymbol{a})$. In other words, the safety function possibly returns that an action is unsafe, which is indeed safe, while it never predicts truly unsafe actions to be safe. Moreover, we define the set of provably safe states based on the safety function: $\mathbb{S}_{\varphi} = \{\boldsymbol{s} | \exists \boldsymbol{a} \in \mathbb{A}, \varphi(\boldsymbol{s}, \boldsymbol{a}) = 1\} \subseteq \mathbb{S}_{\boldsymbol{s}}$. Consequently, for a verified state-action tuple all reachable next states $\boldsymbol{s}'$ are in $\mathbb{S}_{\varphi}$. All provably safe RL approaches rely on the availability of provably safe actions and states to achieve a provably safe system:

**Proposition 1** *Let the system be initiated in a provably safe state $\boldsymbol{s}_0^{\varphi} \in \mathbb{S}_{\varphi}$. Then, there exists a sequence of provably safe actions that ensures $\boldsymbol{s} \in \mathbb{S}_{\boldsymbol{s}}$ at all times until the episode horizon $p$.*

The proof can be easily obtained from the definitions by induction as $\varphi(\boldsymbol{s}, \boldsymbol{a}) = 1$ if $\boldsymbol{s}' \in \mathbb{S}_{\boldsymbol{s}} \wedge \mathbb{A}_{\varphi}(\boldsymbol{s}') \neq \emptyset$. Note that if the episode horizon is finite, the last state of an episode is verified safe if it is contained in $\mathbb{S}_{\boldsymbol{s}}$. A provably safe action must not necessarily exist for this last state.

We define $\mathbb{S}_{\boldsymbol{s}}$ relatively broad since the safety specifications are usually task-specific and can take various forms such as stability, not entering a time-invariant or time-varying unsafe set, and temporal logic specifications. Depending on the safety specification and system under consideration, different verification methods are applicable and encoded in the safety function $\varphi(\boldsymbol{s}, \boldsymbol{a})$. We discuss the concrete verification methods used by previous works in Section 3. Although $\varphi(\boldsymbol{s}, \boldsymbol{a})$ only verifies safety for a given state $\boldsymbol{s}$ and action $\boldsymbol{a}$, it may take more than the next state into account. To make this more graspable, we shortly explain two concepts for

---

[1]The state space is often augmented from the state space for classical control purposes to a state space that includes other safety-relevant dimensions.

[2]Note that "taking no action" is commonly considered to be part of the action space, most often with the action $\boldsymbol{a} = [0, \dots, 0]^{\top}$.

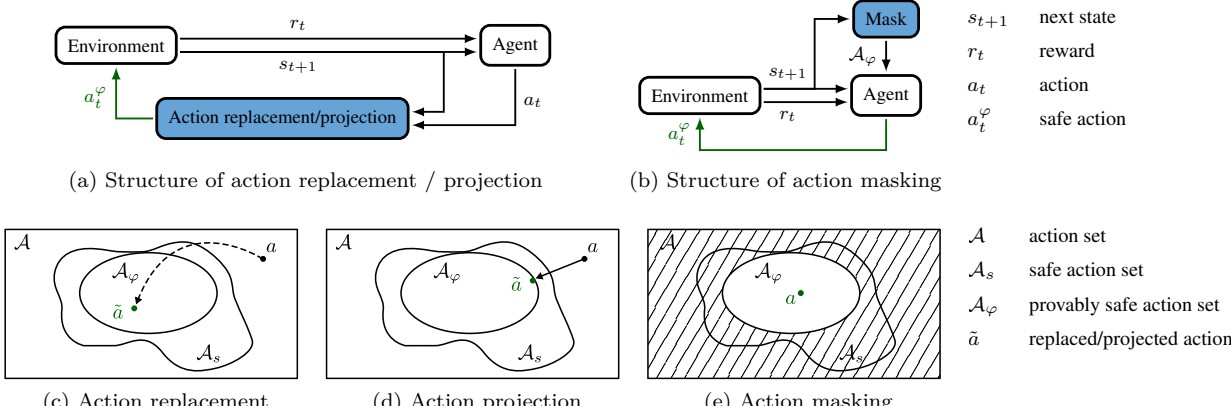

Figure 1: Structure of the three types of provably safe RL methods. The post-posed action replacement or projection methods (a) alter unsafe actions before sending them to the environment. In contrast, preemptive action masking approaches (b) allow the agent to only choose from the safe action space and, therefore, only output safe actions to the environment. Figures (c-e) highlight the differences between the three approaches in the action space. Here, action replacement (c) replaces unsafe actions with actions from the safe action space, action projection (d) projects unsafe actions to the closest safe action, and action masking (e) lets the agent choose solely from the safe action set.

calculating the provably safe states set $\mathbb{S}_\varphi$. For our benchmarks in Section 4, we compute a control invariant set as $\mathbb{S}_\varphi$. Thus, the online verification only consists of checking that all reachable states are contained in this control invariant set. Another concept is predicting the reachable states until a specified time horizon while starting from $s$ and applying $a$ for the first time step and an action sequence for the consecutive time steps. The verification checks that all reachable states are in $\mathbb{S}_s$ and that either the reachable set at the prediction horizon is contained in a safe terminal set or the prediction horizon is the episode horizon. If the verification succeeds, we know that $s \in \mathbb{S}_\varphi$.

Provably safe RL relies on model knowledge to provide safety guarantees, i.e., a conformant model that covers the safety-relevant system and environment dynamics. Hereby, the verification process can use an abstraction of the real system dynamics as long as it is conformant (Roehm et al., 2019; Liu et al., 2023) to the real system, i.e., it over-approximates both aleatoric and epistemic uncertainties and covers all relevant safety aspects. This eases efficient verification, as the complexity of the abstraction is usually significantly lower than the complexity of the underlying MDP. In systems where such a safety model is unavailable, provably safe RL is not applicable, and only non-provably safe approaches, as discussed in Section 1, can be used. In practice, the safety specifications are often weakened to legal or passive safety. Hereby, inevitable safety violations caused by other agents are not considered to be the fault of the agent and are, therefore, not considered unsafe. Examples of proving legal safety have been presented for autonomous driving (Pek et al., 2020) and robotics (Bouraine et al., 2012).

There are multiple ways to ensure provable safety for RL systems, which we summarize in three categories: action replacement, where the safety method replaces all unsafe actions from the agent with safe actions, action projection, which projects unsafe actions onto the safe action space, and action masking, where the agent can only choose actions from the safe action space. We choose this categorization, as it represents the three main approaches found in the literature to modify actions and thereby ensure safety for RL. Action replacement and action projection alter the action after the agent returns it. In contrast, action masking lets the agent exclusively choose from the safe action space. Figure 1 displays the basic concept of these methods. The following subsections describe the concept, mathematical formalization, and practical implications of each approach.

### 2.1 Action replacement

The first approach to ensure the safety of actions is to replace any unsafe action returned by the agent with a safe action before its execution. The first step of action replacement is to evaluate the safety of the suggested action $\boldsymbol{a} \in \mathbb{A}$ using $\varphi(\boldsymbol{s}, \boldsymbol{a})$. If the action sampled from the policy $\boldsymbol{\pi}(\mathbf{a}|\boldsymbol{s})$ is not verified as safe, it is replaced with a provably safe replacement action $\tilde{\boldsymbol{a}} = \boldsymbol{\psi}(\boldsymbol{s})$, where $\boldsymbol{\psi} : \mathbb{S} \rightarrow \mathbb{A}_\varphi$ is called replacement function. Following this procedure, it is guaranteed that only safe actions $\boldsymbol{a}^\varphi$ with

$$\boldsymbol{a}^\varphi = \begin{cases} \boldsymbol{a} \sim \boldsymbol{\pi}(\mathbf{a}|\boldsymbol{s}), & \text{if } \varphi(\boldsymbol{s}, \boldsymbol{a}) = 1 \\ \boldsymbol{\psi}(\boldsymbol{s}), & \text{otherwise} \end{cases} \tag{2}$$

are executed. We discuss how this action replacement alters the MDP in the Appendix and additionally refer the interested reader to Hunt et al. (2021).

There are two general replacement functions found in the literature, *sampling* and *failsafe*. In sampling, the replacement function $\boldsymbol{\psi}_{\text{sample}}(\boldsymbol{s})$ uniformly samples a random action from $\mathbb{A}_\varphi(\boldsymbol{s})$. The other approach is to use a failsafe controller $\boldsymbol{\psi}_{\text{failsafe}}(\boldsymbol{s})$ as replacement action, which could also stem from human feedback. In time-critical and complex scenarios, where building $\mathbb{A}_\varphi(\boldsymbol{s})$ online becomes too time-consuming, $\boldsymbol{\psi}_{\text{failsafe}}(\boldsymbol{s})$ is the only feasible option.

### 2.2 Action projection

In contrast to action replacement, where the replacement action is not necessarily related to the action of the agent, action projection returns the closest provably safe action with respect to the original action and some distance function. For this, we define the optimization problem

$$\tilde{\boldsymbol{a}} = \arg\min_{\hat{\boldsymbol{a}}} \quad \text{dist}\left(\boldsymbol{a}, \hat{\boldsymbol{a}}\right) \tag{3}$$
$$\text{subject to} \quad \varphi(\boldsymbol{s}, \hat{\boldsymbol{a}}) = 1,$$

where $\text{dist}(\cdot)$ describes an arbitrary distance function, e.g., a $p$-norm. Note, that it might not be possible to define such a distance function, especially in discrete action spaces. The constraints are often defined explicitly by $n$ constraint functions $f_i(\tilde{\boldsymbol{a}}, \boldsymbol{s}) \leq 0, \forall i \in 1, \ldots, n$ that confine the next state to the set of provably safe states, i.e., $\boldsymbol{s}' \in \mathbb{S}_\varphi$. The optimization problem in (3) minimizes the alteration of the actions while satisfying the safety specification, which is usually expressed through constraints for the optimization problem. Following Proposition 1, the optimization problem in (3) must always be feasible.

The most prominent ways to formulate the safety constraints for action projection are based on control barrier functions (CBFs) or robust model predictive control (MPC). For the first method, the constraints are defined by CBFs (Wieland & Allgöwer, 2007) that translate state constraints to control input constraints. We formulate the CBFs according to Taylor et al. (2020) as it is an intuitive formulation for RL. Consider a nonlinear control-affine system

$$\dot{\boldsymbol{s}} = \boldsymbol{m}(\boldsymbol{s}) + \boldsymbol{b}(\boldsymbol{s})\,\tilde{\boldsymbol{a}}, \tag{4}$$

where $\boldsymbol{s} \in \mathbb{S} \subseteq \mathbb{R}^N$ is the continuous state with $N$ dimensions, and $\tilde{\boldsymbol{a}} \in \mathbb{A} \subset \mathbb{R}^M$ is the continuous control input with $M$ dimensions and $\boldsymbol{m}(\boldsymbol{s})$ and $\boldsymbol{b}(\boldsymbol{s})$ are locally Lipschitz continuous functions. Then, the function $h$ is a CBF if there exists an extended class $\mathbb{K}$ function $\alpha$ such that (Wabersich et al., 2023, Eq. 10)

$$\nabla h(\boldsymbol{s})(\boldsymbol{m}(\boldsymbol{s}) + \boldsymbol{b}(\boldsymbol{s})\tilde{\boldsymbol{a}}) \geq \alpha(h(\boldsymbol{s})). \tag{5}$$

If the system dynamics are not exactly known, the nominal model can be extended with bounded disturbances $\boldsymbol{d}$ to model the unknown system dynamics:

$$\dot{\boldsymbol{s}} = \boldsymbol{m}(\boldsymbol{s}) + \boldsymbol{b}(\boldsymbol{s})\,\tilde{\boldsymbol{a}} + \boldsymbol{d}. \tag{6}$$

To reduce conservatism, disturbances can be modeled as state and input-depended $\boldsymbol{d}(\boldsymbol{s}, \tilde{\boldsymbol{a}})$, and the maximal occurred disturbance can be learned from data as presented in Taylor et al. (2021). The limitation to control-affine systems makes formulating the constrained optimization problem efficient, e.g., for a Euclidean norm

as the distance function, (3) results in a quadratic program. A downside of using CBFs is that $h(\boldsymbol{s})$ is not trivial to find, especially in environments with dynamic obstacles.

For the second common projection method, we formulate the optimization problem with MPC according to Wabersich & Zeilinger (2021). There, the constraint in (3) is satisfied if we find an action sequence that steers the system from the current state $\boldsymbol{s}$ into the safe terminal set $\mathbb{M}$ within a finite prediction horizon $L \in \mathbb{N}$ while respecting input and state constraints, which reflect the safety specification (Wabersich & Zeilinger, 2021, Eq. 5):

$$\tilde{\boldsymbol{a}} = \underset{\hat{\boldsymbol{a}}}{\arg\min} \quad \text{dist}\,(\boldsymbol{a}, \hat{\boldsymbol{a}}) \tag{7}$$
$$\text{subject to} \quad \boldsymbol{s}_{l+1} = \boldsymbol{g}(\boldsymbol{s}_l, \boldsymbol{a}_l), \boldsymbol{s}_0 = \boldsymbol{s},$$
$$\forall\, l \in \{1, ..., L-1\}\,:\, \boldsymbol{s}_l \in \mathbb{S}_{\boldsymbol{s}},$$
$$\boldsymbol{s}_L \in \mathbb{M},$$
$$\forall\, l \in \{0, ..., L-1\}\,:\, \boldsymbol{a}_l \in \mathbb{A},$$
$$\hat{\boldsymbol{a}} = \boldsymbol{a}_0,$$

where $\boldsymbol{s}_l$ and $\boldsymbol{a}_l$ are the predicted state and action $l$ steps ahead of the current time step.[3] The function $\boldsymbol{g}(\cdot, \cdot)$ is obtained by time discretizing a smooth continuous-time nonlinear system, whose dynamics are governed by $\dot{\boldsymbol{s}} = f(\boldsymbol{s}, \boldsymbol{a})$. The safe terminal set $\mathbb{M} \subseteq \mathbb{S}$ is control invariant, i.e., after the agent has entered $\mathbb{M}$, the associated invariance-enforcing controller keeps the agent inside this set indefinitely. If the optimization problem is solvable, $\tilde{\boldsymbol{a}}$ is executed. If it is not solvable, the control sequence from the previous state is used as a backup plan until the safe terminal set is reached or the optimization problem is solvable again (Schürmann et al., 2018). For perturbed systems of the form $\dot{\boldsymbol{s}} = f(\boldsymbol{s}, \boldsymbol{a}, \boldsymbol{d})$ with a bounded disturbance $\boldsymbol{d}$, robust MPC schemes, e.g., Schürmann et al. (2018), have to be employed and output-feedback MPC schemes, e.g., Gruber & Althoff (2021), account for measurement uncertainties. Similar to the CBF approach, conservatism can be reduced by learning the disturbance bounds from data (Hewing et al., 2020). Note that, for an environment with dynamic obstacles, the safe terminal set can be time-dependent, and we are unaware of a straightforward integration where Proposition 1 still holds.

## 2.3 Action masking

The two previous approaches modify unsafe actions from the agent. In action masking, we do not allow the agent to output an unsafe action in the first place (preemptive method). Hereby, a mask is added to the agent so that it can only choose from actions in the provably safe action set. In addition to Proposition 1, action masking in practice requires an efficient function $\boldsymbol{\eta}(\boldsymbol{s}) : \mathbb{S} \to \mathcal{P}(\mathbb{A})$, where $\mathcal{P}$ denotes the powerset, that determines a sufficiently large set of provably safe actions $\mathbb{A}_\varphi \subseteq \mathbb{A}$ for a given state $\boldsymbol{s}$. The policy function $\boldsymbol{\pi}$ is informed by the function $\boldsymbol{\eta}(\boldsymbol{s})$ and the action selection is adapted such that only actions from $\mathbb{A}_\varphi$ can be selected:

$$\boldsymbol{a} \sim \boldsymbol{\pi}(\mathbf{a}|\boldsymbol{\eta}(\boldsymbol{s}), \boldsymbol{s}) \in \mathbb{A}_\varphi. \tag{8}$$

If $\boldsymbol{\eta}(\boldsymbol{s})$ can only verify one or a few actions efficiently, the agent cannot learn properly because the agent cannot explore different actions and find the optimal one among them. Ideally, the function $\boldsymbol{\eta}(\boldsymbol{s})$ achieves $\mathbb{A}_\varphi = \mathbb{A}_s$.

The action masking approaches for discrete and continuous action spaces are not easily transferable into each other, and will therefore be discussed separately in this subsection. For discrete actions, the safety of each action is typically verified in each state using $\varphi(\boldsymbol{s}, \boldsymbol{a})$ and all verified safe actions are added to $\mathbb{A}_\varphi(\boldsymbol{s})$, i.e., $\boldsymbol{\eta}$ iterates over all actions for the current state $\boldsymbol{s}$ with $\varphi(\boldsymbol{s}, \boldsymbol{a})$ to identify $\mathbb{A}_\varphi(\boldsymbol{s})$. Intuitively, the discrete action mask is an informed drop-out layer added at the end of the policy network. We define the resulting safe policy $\boldsymbol{\pi}_m(\boldsymbol{a}|\boldsymbol{s})$ based on Huang & Ontañón (2022, Eq. 1) as

$$\boldsymbol{\pi}_m(\boldsymbol{a}|\boldsymbol{s}) = \varphi(\boldsymbol{s}, \boldsymbol{a})\, \frac{\boldsymbol{\pi}(\mathbf{a}|\boldsymbol{s})}{\sum_{\boldsymbol{a}^\varphi \in \mathbb{A}_\varphi(\boldsymbol{s})} \boldsymbol{\pi}(\boldsymbol{a}^\varphi|\boldsymbol{s})}. \tag{9}$$

---

[3] We omitted in (7) that $\boldsymbol{s}_1, ..., \boldsymbol{s}_L, \boldsymbol{a}_1, ..., \boldsymbol{a}_{L-1}$ are decision variables of the optimization problem to improve the readability.

The integration of masking in a specific learning algorithm is not trivial. The effects on policy optimization methods are discussed in Krasowski et al. (2020); Huang & Ontañón (2022). For RL algorithms that learn the Q-function, we exemplary discuss the effects of discrete action masking for deep Q-network (DQN) (Mnih et al., 2013), which is most commonly used for Q-learning with discrete actions. During exploration with action masking, the agent samples its actions uniformly from $\mathbb{A}_\varphi$. When the agent exploits the Q-function, it chooses only the best action among $\mathbb{A}_\varphi$, i.e., $\arg\max_{\boldsymbol{a}\in\mathbb{A}_\varphi} Q(\boldsymbol{s}, \boldsymbol{a})$. The temporal difference error for updating the Q-function $Q(\boldsymbol{s}, \boldsymbol{a})$ is (Mnih et al., 2013, Eq. 3)

$$r(\boldsymbol{s}, \boldsymbol{a}) + \gamma \max_{\boldsymbol{a}'} Q(\boldsymbol{s}', \boldsymbol{a}') - Q(\boldsymbol{s}, \boldsymbol{a}), \tag{10}$$

where the action in the next state is $\boldsymbol{a}' \in \mathbb{A}_\varphi$ in contrast to the vanilla temporal difference error where the maximum Q-value for the next state is searched among actions from $\mathbb{A}$. Using the adapted temporal difference error in (10), the learning updates are performed only with Q-values of actions relevant in the next state instead of the full action space.

To comprehensively compare the different provably safe RL approaches on discrete and continuous action spaces in Section 4, we propose a simple formulation for continuous masking since there is no existing approach. We formulate this form of continuous action masking as a transformation of the action of agents to the provable safe action set. Our approach requires both $\mathbb{A}$ and $\mathbb{A}_\varphi$ to be axis-aligned boxes with the same center. We propose to transform the action space $\mathbb{A}$ into $\mathbb{A}_\varphi$ by applying the transformation

$$\tilde{\boldsymbol{a}} = (\boldsymbol{a} - \min(\mathbb{A})) \frac{\max(\mathbb{A}_\varphi) - \min(\mathbb{A}_\varphi)}{\max(\mathbb{A}) - \min(\mathbb{A})} + \min(\mathbb{A}_\varphi) \tag{11}$$

to the actions $\boldsymbol{a} \in \mathbb{A}$, where $\min(\cdot)$ and $\max(\cdot)$ return a vector containing the minimal and maximal value of the given set in each dimension respectively, and all operations are evaluated element-wise. For example, given a two-dimensional continuous action space $\mathbb{A} = [0, 1] \times [-1, 2]$, then $\min(\mathbb{A}) = [0, -1]^\top$. Note that the representation of $\mathbb{A}_\varphi$ as an axis-aligned box centered in $\mathbb{A}$ can be under-approximative and, thus, lead to conservative behavior. To overcome this limitation, more complex set representations, such as the zonotopes (Althoff et al., 2021), for the action spaces ($\mathbb{A}$ and $\mathbb{A}_\varphi$) in combination with solving an optimization problem that maximizes the size of $\mathbb{A}_\varphi$ could be investigated. A less sophisticated yet possibly effective approach is searching for a good latent interval representation of and transformation to $\mathbb{A}_\varphi$ by applying principal component analysis to a set of $\mathbb{A}_\varphi$ for different states as a pre-computing step. Since the action spaces for RL are defined a priori, there must always be a valid transformation from $\boldsymbol{a}$ to $\tilde{\boldsymbol{a}} \in \mathbb{A}_\varphi$ and such that the operation is time-invariant for all state-action pairs. In the next section, we discuss the effect of the three provably safe RL approaches on the policy distribution and exploration.

## 2.4 Impact on the distribution of actions

The three previously presented provably safe RL methods have different effects on the resulting distribution of actions, as illustrated for a one-dimensional continuous action space and a probabilistic policy in Figure 2. For action projection, all actions that are not verified safe $\boldsymbol{a} \notin \mathbb{A}_\varphi$ are projected to the boundary of the provably safe action set $\partial\mathbb{A}_\varphi$. Therefore, actions on $\partial\mathbb{A}_\varphi$ are disproportionately explored compared to the interior of $\mathbb{A}_\varphi$. A similar effect can occur in action replacement depending on the replacement strategy, e.g., with $\boldsymbol{\psi}_{\text{failsafe}}(\boldsymbol{s})$, the failsafe action is explored more often. However, if the random sampling strategy $\boldsymbol{\psi}_{\text{sample}}(\boldsymbol{s})$ is used, as shown in Figure 2 (a), the likelihood of all safe actions being explored increases equally. The sampling strategy, therefore, fosters exploration and discourages exploitation, as all non-provably safe actions lead to uniformly distributed exploration in the safe action space. The distribution of actions for both action replacement and projection differs from the distribution of the current policy, which might be problematic for on-policy algorithms. In action masking, we only map the exploration from $\mathbb{A}$ to $\mathbb{A}_\varphi$. Thus, the exploration strategy is not affected by action masking. In this aspect, our approach in (11) is conceptually similar to action normalization, which is commonly used in RL (Sutton & Barto, 2018, Ch. 16.8).

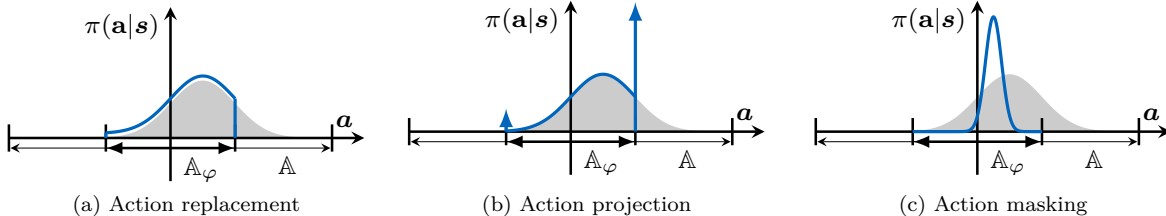

(a) Action replacement       (b) Action projection       (c) Action masking

Figure 2: Idealized impact of the provably safe RL methods on the probability density function of the RL policy for a single action in a given state. The probability density function of the original RL policy $\pi\,(\mathbf{a}|\boldsymbol{s})$ and the provably safe policy $\pi\,(\tilde{\mathbf{a}}|\boldsymbol{s})$ are depicted as the gray area and the blue line respectively. Figure (a) shows action replacement with the random sampling strategy. Figure (b) displays action projection, where the vertical arrows are scaled Dirac delta distributions that stem from the fact that the unsafe parts of the original policy distribution are projected to the boundary of $\mathbb{A}_\varphi$. Figure (c) depicts our proposed implementation of continuous action masking.

## 2.5 Learning tuples

When changing the RL action, the training of the agent can be conducted with four possible learning tuples:

- *naive* - learning based on the action $\boldsymbol{a}$ returned by the policy network of the agent and the reward $r(\boldsymbol{s}, \boldsymbol{a}^\varphi)$ corresponding to the executed action $\boldsymbol{a}^\varphi$, which we denote by the tuple $(\boldsymbol{s}, \boldsymbol{a}, \boldsymbol{s}', r(\boldsymbol{s}, \boldsymbol{a}^\varphi))$. This ensures that the agent is updated according to its current policy. Learning with the original action $\boldsymbol{a}$ should benefit on-policy learning, where the policy is updated based on experience collected using the most recent policy.

- *adaption penalty* - is *naive* with a penalty $r^*(\boldsymbol{s}, \boldsymbol{a}, \boldsymbol{a}^\varphi) = r(\boldsymbol{s}, \boldsymbol{a}^\varphi) + r_{\mathrm{penalty}}(\boldsymbol{s}, \boldsymbol{a}, \boldsymbol{a}^\varphi)$ if an unsafe action was selected, which we denote by the tuple $(\boldsymbol{s}, \boldsymbol{a}, \boldsymbol{s}', r^*(\boldsymbol{s}, \boldsymbol{a}, \boldsymbol{a}^\varphi))$. In action projection, the penalty $r_{\mathrm{penalty}}$ can include a term that is proportional to the projection distance $\mathrm{dist}(\boldsymbol{a}, \boldsymbol{a}^\varphi)$, as proposed by Wang (2022).

- *safe action* - learning based on the safe and possibly adapted action and the corresponding reward, denoted by the tuple $(\boldsymbol{s}, \boldsymbol{a}^\varphi, \boldsymbol{s}', r(\boldsymbol{s}, \boldsymbol{a}^\varphi))$. By using the *safe action* tuple, we are correctly rewarding the agent for the actually performed transition. However, this requires updating the agent with an action that did not stem from its current policy $\boldsymbol{\pi}(\mathbf{a}|\boldsymbol{s})$, which is an expected behavior in off-policy but not on-policy learning. So, the safe action tuple might be a better fit for off-policy than for on-policy RL.

- *both* - in case the RL agent proposes an unsafe action, both the *adaption penalty* and the *safe action* tuples are used for learning.

In all cases, the next state $\boldsymbol{s}'$ and reward $r$ are the true state and reward received from the environment after executing the safe action $\boldsymbol{a}^\varphi$.

Action masking is always paired with the *naive* or *adaption penalty* learning tuple in the literature. The *adaption penalty* is related to the reduction of the action space due to masking or a safety component in the reward function, and not a sparse reward as for action projection and action replacement. For discrete action masking, the naive and the *safe action* tuples are equivalent since the agent is only allowed to choose provably safe actions (see Figure 1). For continuous action masking, using the *safe action* tuple with the transformed action $\tilde{\boldsymbol{a}}$ leads to inconsistencies in learning because every action is transformed if $\mathbb{A}_\varphi \neq \mathbb{A}$.

## 3 Literature review

In this section, we summarize previous works in provably safe RL and assign them to the proposed categories. To identify the related literature, we used the search string `TITLE-ABS("reinforcement learning") AND TITLE(learning) AND [TITLE(safe*) OR TITLE(verif*) OR TITLE(formal*) OR TITLE(shield*)] AND LIMIT-TO(LANGUAGE,"English")`[4] for the Scopus[5] and IEEEXplore[6] search engine, which led to 620 papers already removing duplicates. Then, we screened papers by title and abstracts to identify 160 seemingly relevant papers. After close inspection, we identified 47 of these 160 papers as provably safe RL works. We give a condensed overview of all application-independent provably safe RL works in Table 1, and cluster all 47 provably safe RL works in Table 2 by their application. Some approaches in Table 1 are presented for unbounded disturbance. In such a setting, hard safety guarantees are generally not achievable. Still, we include approaches that would be provably safe with the assumption that the disturbance is bounded.

**Action replacement**  One of the earliest provably safe RL works is Alshiekh et al. (2018), which constructed a so-called safety shield from linear temporal logic formulas. For that, they construct the verification function $\varphi(\boldsymbol{s}, \boldsymbol{a})$ by converting the linear temporal logic formulas into an automaton, on which they perform model checking. The advantage of online model checking is that linear temporal logic constraints can be guaranteed for general nonlinear systems, and the online complexity is linear in the number of discrete states. However, the method is only applicable to small discrete state spaces, as constructing the automaton offline has exponential complexity in the number of discrete states, and the online checking complexity also increases exponentially with the formula length (Baier & Katoen, 2008). In Alshiekh et al. (2018), the agent outputs $n$ ranked actions, which are all checked for safety using $\varphi(\boldsymbol{s}, \boldsymbol{a})$. The shield executes the highest-ranked safe action or replaces the action with $\boldsymbol{\psi}_{\mathrm{sample}}(\boldsymbol{s})$ if none of the $n$ actions is safe. They update the agent with the *safe action* learning tuple but also propose that the *both* learning tuple can be used to obtain additional training information. Similarly to Alshiekh et al. (2018), Könighofer et al. (2020) show that both probabilistic and deterministic shields increase the sample efficiency and performance for both action replacement and masking methods.

Akametalu et al. (2014) propose action replacement based on Hamilton-Jacobi-Isaacs reachability analysis, which was later extended by Fisac et al. (2019) to a general safe RL framework. They determine $\mathbb{S}_{\varphi}$ using Hamilton-Jacobi-Isaacs reachability analysis given bounded system disturbances. Safety is guaranteed by replacing any learned action on the border of $\mathbb{S}_{\varphi}$ with $\boldsymbol{\psi}_{\mathrm{failsafe}}(\boldsymbol{s})$ stemming from an Hamilton-Jacobi-Isaacs optimal controller to guide the system back inside the safe set. Hamilton-Jacobi-Isaacs reachability analysis can verify reach-avoid problems with arbitrary non-convex sets (Wabersich et al., 2023) and disturbances that stem from a compact set. However, constructing the safe set scales exponentially in complexity with the number of state dimensions, which makes Hamilton-Jacobi-Isaacs reachability analysis infeasible for systems with more than four state dimensions (Chen & Tomlin, 2018). Fisac et al. (2019) argue that replacing the unsafe action with the action that maximizes the distance to the unsafe set increases performance in uncertain real-world environments compared to action projection. However, Hamilton-Jacobi-Isaacs approaches suffer from the curse of dimensionality and are, therefore, only feasible for systems with specific characteristics (Herbert et al., 2021).

Shao et al. (2021) use a trajectory safeguard based on set-based reachability analysis for $\varphi(\boldsymbol{s}, \boldsymbol{a})$. Set-based reachability analysis is applicable to reach-avoid problems for general nonlinear systems with uncertainties in the initial state, system dynamics, and input disturbances as long as they stem from a compact set, see, e.g., Althoff et al. (2021). Set-based reachability analysis has polynomial complexity in the state dimension for most set representations, as discussed by Althoff et al. (2021). However, compared to Hamilton-Jacobi-Isaacs reachability analysis, set-based reachability analysis cannot handle arbitrary non-convex sets but depends on specific set representations. Shao et al. (2021) sample $n$ new actions randomly in the vicinity of the action if the action is unsafe. They then execute the closest safe action to the original (unsafe) action. If none of the $n$ new actions is safe, a failsafe action $\boldsymbol{\psi}_{\mathrm{failsafe}}(\boldsymbol{s})$ is executed. Shao et al. (2021) train their agent

---

[4]Documentation at http://schema.elsevier.com/dtds/document/bkapi/search/SCOPUSSearchTips.htm
[5]scopus.com
[6]ieeexplore.ieee.org

Table 1: Comparison of application-independent provably safe RL approaches.

| Reference | Verification Method | Space | | Learning Tuple | Environment[1] |
|---|---|---|---|---|---|
| | | State | Action | | |
| **Action replacement** | | | | | |
| Akametalu et al. (2014) | HJI[2] reachability analysis | cont. | cont. | special RL alg. | 1D quadrotor [stoch.], cart-pole [stoch.] |
| Fisac et al. (2019) | HJI reachability analysis | cont. | cont. | N/A | 1D quadrotor [stoch.] |
| Alshiekh et al. (2018) | model checking of automaton constructed from LTL[3] | disc. | disc. | *safe action* | Grid world [stoch.] |
| Könighofer et al. (2020) | model checking of automaton constructed from LTL | disc. | disc. | *adaption penalty* | ACC[4] [stoch.] |
| Anderson et al. (2020) | robust control invariant set | cont. | cont. | N/A | pendulum [det.], reach-avoid [det.], others [det.] |
| Hunt et al. (2021) | theorem proving of $\mathrm{d}\mathcal{L}$[5] formulas | disc. | disc. | *naive* | Grid world [stoch.] |
| Bastani (2021) | MPC[6] | cont. | cont. | deployment only | bicycle [det.], cart-pole [det.] |
| Shao et al. (2021) | set-based reachability analysis | cont. | cont. | *naive* | 3D quadrotor [det.], highway driving [det.] |
| Selim et al. (2022b) | set-based reachability analysis | cont. | cont. | *adaption penalty* | 3D quadrotor [stoch.], mobile robot [stoch.] |
| **Action projection** | | | | | |
| Pham et al. (2018) | verification of affine constraints for actions | cont. | cont. | *adaption penalty* | manipulator [det.] |
| Cheng et al. (2019) | CBF[7] | cont. | cont. | *safe action* | ACC [stoch.], pendulum [stoch.] |
| Li et al. (2019a) | CBF synthesized from LTL | cont. | cont. | *adaption penalty* | manipulator [det.] |
| Gros et al. (2020) | MPC | cont. | cont. | *naive* | 2D LTI[8] system [stoch.] |
| Wabersich & Zeilinger (2021) | MPC | cont. | cont. | *naive* | 3D quadrotor [stoch.], pendulum [stoch.] |
| Marvi & Kiumarsi (2022) | CBF | cont. | cont. | *adaption penalty* | 2D LTI system [det.] |
| Selim et al. (2022a) | set-based reachability analysis | cont. | cont. | *naive* | 3D quadrotor [stoch.], mobile robot [stoch.] |
| Kochdumper et al. (2023) | set-based reachability analysis | cont. | cont. | *adaption penalty* | 3D quadrotor [stoch.] |
| **Action masking** | | | | | |
| Fulton & Platzer (2018) | theorem proving of $\mathrm{d}\mathcal{L}$ formulas | cont. | disc. | *naive* | ACC [det.] |
| Fulton & Platzer (2019) | theorem proving of $\mathrm{d}\mathcal{L}$ formulas | cont. | disc. | *naive* | ACC [stoch.] |
| Huang & Ontañón (2022) | verification of affine equality constraints for actions | disc. | disc. | *naive* | Grid world [N/A] |
| This study | set-based reachability analysis | cont. | cont. | *naive* | pendulum [det.], 2D quadrotor [stoch.] |

Abbreviations: [1]stoch.: stochastic environment model, det.: deterministic environment model, [2]Hamilton-Jacobi-Isaacs (HJI), [3]linear temporal logic (LTL), [4]adaptive cruise control (ACC), [5]differential dynamic logic ($\mathrm{d}\mathcal{L}$), [6]model predictive control (MPC), [7]control barrier function (CBF), [8]linear time-invariant (LTI).

on the *naive* learning tuple. Selim et al. (2022b) also verify the safety of actions with set-based reachability analysis. They propose an informed replacement for $\boldsymbol{\psi}(\boldsymbol{s})$ such that the reachable set of the controlled system is pushed away from the unsafe set $\mathbb{S} \setminus \mathbb{S}_{\boldsymbol{s}}$. The authors further propose a method to account for unknown system dynamics using a so-called black-box reachability analysis. They use the *adaption penalty* learning tuple and showcase that their approach achieves provable safety in three use cases.

Hunt et al. (2021) build the verification function $\varphi(\boldsymbol{s}, \boldsymbol{a})$ using theorem proving of differential dynamic logic formulas. Using $\varphi(\boldsymbol{s}, \boldsymbol{a})$, they determine $\mathbb{A}_{\varphi}(\boldsymbol{s})$ for discrete action spaces and use $\boldsymbol{\psi}_{\mathrm{failsafe}}(\boldsymbol{s})$ for replacement. They further show how provably safe end-to-end learning can be accomplished using controller and model monitors. They train the RL agent using the *naive* learning tuple on a drone example. The work of Anderson et al. (2020) proposes to define $\mathbb{S}_{\varphi}$ as a robust control invariant set and construct the safety function $\varphi(\boldsymbol{s}, \boldsymbol{a})$ based on a worst-case linear model of the system dynamics. A further notable work is Bastani (2021), which proposes a model predictive shield alongside the trained policy, which uses $\boldsymbol{\psi}_{\mathrm{failsafe}}(\boldsymbol{s})$ for action replacement.

Table 2: Overview of applications in provably safe RL.

| | Action Replacement | Action Projection | Action Masking |
|---|---|---|---|
| Aerial Vehicles | [‡‡]Akametalu et al. (2014); Shyamsundar et al. (2017); [‡‡]Fisac et al. (2019); Anderson et al. (2020); [†]Harris & Schaub (2020); [†]Shao et al. (2021); [†]Selim et al. (2022b); [†]Nazmy et al. (2022) | [†]Wabersich & Zeilinger (2021); [†]Selim et al. (2022a); [†]Kochdumper et al. (2023) | N/A |
| Autonomous Driving | [†]Chen et al. (2020); Könighofer et al. (2020); [†]Shao et al. (2021); [†]Chen et al. (2022a); [†]Lee & Kwon (2022); [‡‡]Wang et al. (2023); [†]Evans et al. (2023) | Cheng et al. (2019); [†]Wang (2022); [†]Hailemichael et al. (2022b); [†]Hailemichael et al. (2022a); [‡‡]Kochdumper et al. (2023) | Fulton & Platzer (2018); [†]Mirchevska et al. (2018); Fulton & Platzer (2019); [†]Krasowski et al. (2020); Brosowsky et al. (2021); [†]Krasowski et al. (2022) |
| Power Systems | [†]Ceusters et al. (2023) | [†]Eichelbeck et al. (2022); [†]Chen et al. (2022b); [†]Zhang et al. (2023); [†]Yu et al. (2023) | [†]Tabas & Zhang (2022) |
| Robotic Manipulation | [†]Thumm & Althoff (2022) | [‡‡]Pham et al. (2018); [‡‡]Li et al. (2019a) | N/A |
| Control Benchmarks | Akametalu et al. (2014); Anderson et al. (2020); Bastani (2021); Shao et al. (2021) | Cheng et al. (2019); Gros et al. (2020); Wabersich & Zeilinger (2021); Marvi & Kiumarsi (2022) | N/A |
| Grid World Games | Alshiekh et al. (2018); Hunt et al. (2021) | N/A | Huang & Ontañón (2022) |
| Miscellaneous | *Active suspension*: Li et al. (2019b); *Computing networks*: [†]Wang et al. (2022); *Mobile robot*: [†]Selim et al. (2022b) | *Mobile robot*: [†]Selim et al. (2022a); *Engine emission*: [†]Norouzi et al. (2023) | *Computing networks*: Seetanadi et al. (2020); *Traffic signal*: [†]Müller & Sabatelli (2022) |

Note: Studies using high-fidelity simulators are marked with [†], and [‡‡] indicates physical experiments. Additionally, papers occur multiple times in case they demonstrate their approach for different applications.

The most popular method by publications is action replacement. This is also visible from the large variety of application-specific approaches, e.g., for aerial vehicles (Shyamsundar et al., 2017; Harris & Schaub, 2020; Nazmy et al., 2022), autonomous driving (Chen et al., 2020; 2022a; Lee & Kwon, 2022; Wang et al., 2023; Evans et al., 2023), power systems (Ceusters et al., 2023), robotic manipulation (Thumm & Althoff, 2022), active suspension systems (Li et al., 2019b), and traffic engineering in computing networks (Wang et al., 2022). In particular, Ceusters et al. (2023) compare fail-safe action replacement and sampling-based action replacement. They observe that both methods have higher initial performance than the unsafe RL baseline, and fail-safe action replacement leads to better performance than the sampling-based version.

**Action projection**  Research on action projection is usually conducted on continuous action and state spaces. The main differentiating factor between studies in this category is the specification of the optimization problem for the projection. To begin with, the work of Pham et al. (2018) guarantees safety using a differentiable constrained optimization layer called OptLayer. Their approach is restricted to quadratic programming problems, so the system model and constraints have to be linear. Despite these limitations, they show the effectiveness of their approach on a collision avoidance task with a simple robotic manipulator.

Cheng et al. (2019) specify the safety constraint $\varphi(\boldsymbol{s}, \boldsymbol{a}) = 1$ of the optimization problem via CBFs. Thus, the optimization problem in (3) becomes a quadratic program. Theoretically, the CBF approach is applicable to general control-affine systems with disturbances from a compact set for reach-avoid specifications. However, finding a CBF is not trivial, and synthesizing them can be exponential in the system dimension (Ames et al., 2019). To solve (3) more efficiently online, Cheng et al. (2019) add a neural network to the approach

in Section 2.2, which approximates the correction due to the CBF. The action is then shifted by the approximated value prior to optimization. This shift improves the implementation efficiency while still guaranteeing safety, as the action is often already safe after the shift, and no optimization problem needs to be solved. Their safe learning with CBFs shows faster convergence speed than vanilla RL when learning on a pendulum and a car following task. Li et al. (2019a) propose a method to construct a continuous CBF from an automaton, which is defined by linear temporal logic formulas. They further construct a guiding reward from the given automaton to improve the learning performance. The proposed approach is capable of learning a high-dimensional cooperative manipulation task safely. The authors of Marvi & Kiumarsi (2022) define a different problem, where the system model is assumed to be deterministic but unknown. They learn an optimal controller and the system dynamics iteratively while decreasing the conservativeness of their CBF in each iteration. The approach is provably safe for linear time-invariant (LTI) systems without disturbances.

Gros et al. (2020) and Wabersich & Zeilinger (2021) implement the optimization problem as a robust MPC problem as defined in (7). MPC is applicable to reach-avoid problems with measurement and state disturbances. However, when controlling high-speed systems, robust MPC is often limited to linear systems (Zeilinger et al., 2014). Gros et al. (2020) mainly discuss how the learning update has to be adapted if action projection is used for different RL algorithms. For Q-learning, they find that no adaption of the learning algorithm is necessary when the *naive* tuple is used. For policy gradient methods, they argue that the projection must also be included in the gradient for stable learning (Gros et al., 2020, Sec. 3). One downside of the robust MPC formulation of Gros et al. (2020) and Wabersich & Zeilinger (2021) is that dynamic constraints originating from moving obstacles or persons in the environment are not trivial to integrate. They approximate the dynamics of the system with a Gaussian process (GP) so that hard safety guarantees are impossible to prove. However, they could guarantee hard safety specifications if they would assume deterministic system dynamics with bounded disturbance as the aforementioned approaches do. Gros et al. (2020) evaluate their approach on a simple 2D LTI system, and Wabersich & Zeilinger (2021) show the efficacy of their approach on a pendulum and a quadrotor task.

Contrary to their previous work in Selim et al. (2022b), Selim et al. (2022a) propose to solve an optimization problem to find the closest safe action instead of using an informed replacement. They again use set-based reachability analysis to construct $\varphi(\boldsymbol{s}, \boldsymbol{a})$. They test their approach on a quadrotor and mobile robot benchmark. Kochdumper et al. (2023) utilize set-based reachability analysis to verify actions in $\varphi(\boldsymbol{s}, \boldsymbol{a})$. They formulate the projection for a parameterization of the action space and arrive at a mixed-integer quadratic problem with polynomial constraints. Their approach achieves provable safety for nonlinear systems with bounded disturbances, and they demonstrate their approach on two quadrotor tasks, autonomous driving on highways, and a physical F1TENTH car.

Next to the conceptual approaches, action projection algorithms are also specifically proposed for many cyber-physical systems, such as autonomous driving (Wang, 2022; Hailemichael et al., 2022a;b), power systems (Eichelbeck et al., 2022; Chen et al., 2022b; Zhang et al., 2023; Yu et al., 2023), and engine emission control (Norouzi et al., 2023). Wang (2022) compares the deployment of a discrete action masking approach with her continuous action projection approach. The goal-reaching performance is lower for the discrete action masking approach. However, this could be due to the coarse discretization of the action space in three actions.

**Action masking**   To the best of our knowledge, the existing literature considers action masking only for discrete action spaces. The work Huang & Ontañón (2022) analyzes the effect of discrete action masking on the policy gradient algorithm in RL, but they assume that $\mathbb{A}_s$ is known, which is typically only the case in game and grid world environments.

The two main works investigating action masking are Fulton & Platzer (2018) and Fulton & Platzer (2019). They construct controller and model monitors based on theorem proving of differential dynamic logic specifications, see Platzer (2008). The controller monitor is used to build the mask $\boldsymbol{\eta}(\boldsymbol{s})$, and the model monitor verifies if the underlying system model is correct based on previous transitions. In each state, the agent can choose from the set of actions that the controller monitor verified as safe. Identifying the correct system can be challenging, thus an approach to automatically generate candidates is introduced as well. Their approach

is provably safe if the initial model is correct (Fulton & Platzer, 2018) or multiple models are given, from which at least one is correct (Fulton & Platzer, 2019) for all times. They validate their provably safe action masking on adaptive cruise control tasks.

In addition to the works mentioned above, there are works investigating action masking for the specific application of autonomous driving (Mirchevska et al., 2018; Krasowski et al., 2020; Brosowsky et al., 2021; Krasowski et al., 2022), power systems (Tabas & Zhang, 2022), adaptive routing in computing networks (Seetanadi et al., 2020), and urban traffic signal control (Müller & Sabatelli, 2022). The only application-specific approach that compares action masking with other provably safe RL approaches is Brosowsky et al. (2021). They observe that their masking approach converges slightly faster than action projection.

## 4 Experimental comparison

In this section, we evaluate the performance of the three provably safe RL classes and the four learning tuples introduced in Section 2. For our comparison, we select an inverted pendulum and a 2D quadrotor stabilization task[7], as these benchmarks are commonly evaluated in related works presented in Table 1. The provably safe state set $\mathbb{S}_\varphi$ is the same for all three approaches and, therefore, comparable. We add system disturbances to the benchmarks to make them more realistic and show that the provably safe RL approaches can handle disturbances sampled from a compact disturbance set. Despite their low dimensionality, our results are likely transferable to real-world systems since real-world complexity is often reduced in practice by using lower-dimensional abstract models and an additional disturbance term. Conformance checking techniques (Roehm et al., 2019; Liu et al., 2023) can then guarantee that the abstract model incorporates recorded real-world behaviors of the system.

The algorithms shown in this section are action replacement with $\psi_{\text{sample}}(s)$, action projection using affine constraints, and action masking. We compare each configuration on ten random seeds and five common RL algorithms[8]: continuous Twin Delayed Deep Deterministic policy gradient algorithm (TD3) (Fujimoto et al., 2018), continuous soft actor-critic (SAC) (Haarnoja et al., 2018), discrete DQN (Mnih et al., 2013), and continuous and discrete proximal policy optimization (PPO) (Schulman et al., 2017).

### 4.1 Environments

We compare the provably safe RL approaches on an inverted pendulum and a 2D quadrotor stabilization task.

**Inverted pendulum**  The state of the pendulum is defined as $s = [\theta, \dot{\theta}]^\top$, and follows the dynamics

$$\dot{s} = \begin{pmatrix} \dot{\theta} \\ \frac{g}{l}\sin(\theta) + \frac{1}{ml^2}a \end{pmatrix}, \tag{12}$$

where $a$ is the one-dimensional action, $g$ is gravity, $m$ is the mass of the pendulum, $l$ its length, and friction and damping are ignored. We discretize the dynamics using the explicit Euler method. The actions are bounded by $|a| \leq 30\text{rad s}^{-1}$. The desired equilibrium state is $s^* = [0,0]^\top$. The observation and reward are identical to the *OpenAI Gym Pendulum-V0*[9] environment.

**2D quadrotor**  The quadrotor in our experiments can only fly in the $x$-$z$-plane and rotate around the $y$-axis with angle $\theta$. The state of the system is defined as $s = [x, z, \dot{x}, \dot{z}, \theta, \dot{\theta}]^\top$ and the action as $a = [a_1, a_2]^\top$.

---

[7]Our implementation is available at CodeOcean: doi.org/10.24433/CO.9209121.v1 .

[8]All implementations are based on `stable-baselines3` (Raffin et al., 2021).

[9]Available at: gymnasium.farama.org/environments/classic_control/pendulum/

The system dynamics

$$
\dot{\boldsymbol{s}} = \begin{pmatrix} \dot{x} \\ \dot{z} \\ a_1 k \sin(\theta) \\ -g + a_1 k \cos(\theta) \\ \dot{\theta} \\ -d_0\theta - d_1\dot{\theta} + n_0 a_2 \end{pmatrix} + \begin{pmatrix} 0 \\ 0 \\ w_1 \\ w_2 \\ 0 \\ 0 \end{pmatrix} \tag{13}
$$

are based on Mitchell et al. (2019), where $w_1, w_2$ represent the system disturbance, and $k$, $d_0$, $d_1$, and $n_0$ are constant parameters (see Table 4). We linearize the dynamics using a first-order Taylor expansion at the equilibrium point $\boldsymbol{s}^* = [0, 1, 0, 0, 0, 0]^\top$ and obtain the discrete-time system for the linearized dynamics. We sample the disturbance $\boldsymbol{w} = [w_1, w_2]^\top$ uniformly, independent, and identically distributed from a compact disturbance set $\mathbb{W} \subset \mathbb{R}^2$. The actions range from $\boldsymbol{a}_{\min} = \left[-1.5 + \frac{g}{K}, -\frac{\pi}{12}\right]^\top$ to $\boldsymbol{a}_{\max} = \left[1.5 + \frac{g}{K}, \frac{\pi}{12}\right]^\top$. The reward is defined as $r(\boldsymbol{s}, \boldsymbol{a}) = \exp\left(-\|\boldsymbol{s} - \boldsymbol{s}^*\|_2 - \frac{0.01}{2}\|\frac{\boldsymbol{a} - \min(\mathbb{A})}{\max(\mathbb{A}) - \min(\mathbb{A})}\|_1\right)$.

## 4.2 Computation of the safe state set

To obtain a possibly large set of provably safe states $\mathbb{S}_\varphi$ and a provably safe controller for our environments, we use the scalable approach for computing robust control invariant sets of nonlinear systems presented in Schäfer et al. (2023): for every state $\boldsymbol{s}_0 \in \mathbb{S}_\varphi \subset \mathbb{S}_{\boldsymbol{s}}$, there exists a provably safe action $\tilde{\boldsymbol{a}}_0 \in \mathbb{A}$ so that $\boldsymbol{s}_1 = \boldsymbol{g}(\boldsymbol{s}_0, \tilde{\boldsymbol{a}}_0, \boldsymbol{w}_0) \in \mathbb{S}_\varphi$ with a bounded disturbance $\boldsymbol{w}_0 \in \mathbb{W} \subset \mathbb{R}^O$ where $\mathbb{W}$ has $O$ dimensions. Hence, $\varphi(\boldsymbol{s}_0, \tilde{\boldsymbol{a}}_0) = 1$ for every $\boldsymbol{s}_0 \in \mathbb{S}_\varphi$. In this work, we assume the disturbance to be constant in between sampling times. Note that we use the obtained provable safe controller for the failsafe replacement function $\boldsymbol{\psi}_{\text{failsafe}}(\boldsymbol{s})$. The algorithm in Schäfer et al. (2023) provides an explicit representation of $\mathbb{S}_\varphi$, which enables a fair comparison of our provably safe RL implementations. To retrieve $\mathbb{A}_\varphi$ from $\mathbb{S}_\varphi$ at a given state $\boldsymbol{s}$, we first convert $\mathbb{S}_\varphi$ from generator representation, which is used in Schäfer et al. (2023), into halfspace representation, i.e., $\mathbb{S}_\varphi = \{\boldsymbol{s} | \boldsymbol{C}\boldsymbol{s} \leq \boldsymbol{q}\}$, using the open-source toolbox CORA (Althoff, 2015). We evaluate the safety function given the state $\boldsymbol{s}_k \in \mathbb{S}_\varphi$ and an action $\boldsymbol{a}_k \in \mathbb{A}$ by computing the reachable set $\mathcal{R}(k+1)$ at the next time step, which encloses the states that are reachable for all $\boldsymbol{w}_k \in \mathbb{W}$ (Althoff, 2015). The reachable set can be represented as a zonotope, i.e., $\mathcal{R}(k+1) = \{\boldsymbol{s}_{k+1} | \boldsymbol{s}_{k+1} = \boldsymbol{c} + \boldsymbol{G}\boldsymbol{\beta}, |\boldsymbol{\beta}|_\infty \leq 1\}$. If $\mathcal{R}(k+1) \subseteq \mathbb{S}_\varphi$, the action $\boldsymbol{a}$ is verified as safe, i.e., $\varphi(\boldsymbol{s}_{k+1}, \boldsymbol{a}_{k+1}) = 1$, which holds if and only if (Schürmann et al., 2020, Theorem 2)

$$
\boldsymbol{C}\boldsymbol{c} + |\boldsymbol{C}\boldsymbol{G}|\boldsymbol{1} \leq \boldsymbol{q}, \tag{14}
$$

where the absolute value is applied elementwise and $\boldsymbol{1}$ denotes a vector full of ones of appropriate dimension. The approach of Schäfer et al. (2023) allows us to compute $\mathbb{S}_\varphi$ for high-dimensional nonlinear systems. However, the conversion to halfspace representation is computationally too expensive for higher dimensional systems. Therefore, we plan to develop generator-based versions of our provably safe RL methods in future work to mitigate this shortcoming.

## 4.3 Results

The 2D quadrotor task is the main comparison environment in this work as it is more complex and shows the differences between provably safe RL approaches clearer than the inverted pendulum task. We evaluate the differences between the provably safe RL algorithms in Figure 3 and the effect of different learning tuples in Figure 4. All training runs on all individual algorithms and environments are presented in the Appendix, including a comparison between the $\boldsymbol{\psi}_{\text{sample}}(\boldsymbol{s})$ and $\boldsymbol{\psi}_{\text{failsafe}}(\boldsymbol{s})$ replacement function.

**Comparison of provably safe RL algorithms** The safety violation evaluation of the baselines in Figure 3d shows that the baseline algorithms fail to guarantee safety during training in the 2D quadrotor stabilization task. All provably safe RL algorithms guarantee safety as expected. Between the baselines, TD3 converges significantly faster than all other algorithms.

Figures 3a and 3b show the performance of the three provably safe RL categories (a) action replacement using $\boldsymbol{\psi}_{\text{sample}}(\boldsymbol{s})$, (b) action projection, and (c) action masking together, with the baselines averaged over all

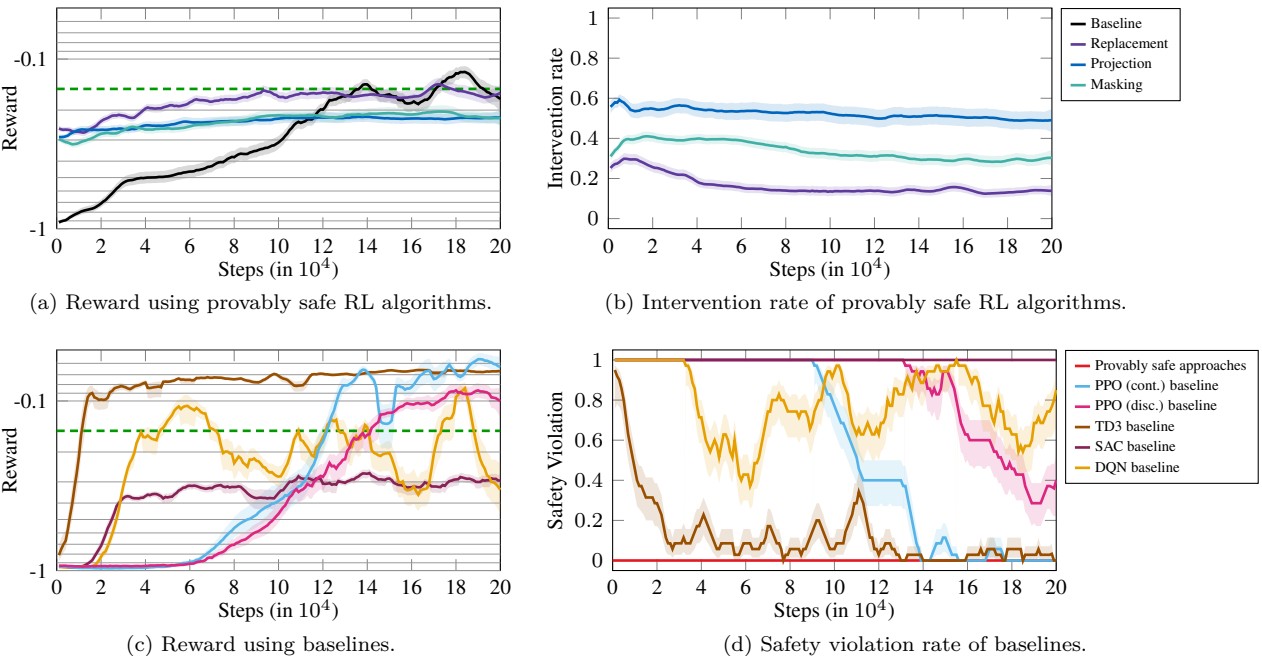

(a) Reward using provably safe RL algorithms.

(b) Intervention rate of provably safe RL algorithms.

(c) Reward using baselines.

(d) Safety violation rate of baselines.

Figure 3: Training curves for the 2D quadrotor benchmark. Top: Comparison of the three provably safe RL classes and the unsafe benchmarks averaged over all algorithms trained with the *naive* tuple. Bottom: Comparison of benchmark algorithms TD3, SAC, DQN, continuous and discrete PPO. All training runs were conducted on ten random seeds per algorithm. The left column depicts the reward. The right column shows the safety violations for the baselines, and the safety intervention rate for the provably safe RL algorithms.

five RL implementations and trained using the *naive* learning tuple. For a better comparison of the reward curves in Figure 3a, we added a dashed green line that indicates the final training reward averaged over all five RL baselines and ten random seeds. The reward comparison shows that action replacement performs better than action projection and masking on average.

We also compare the intervention rate of the three safety mechanisms. For action replacement and projection, our intervention rate metric indicates the share of RL steps per episode in which the safety function altered the action. For action masking, the intervention rate compares the average volume of the provably safe action set over an episode with the volume of the provably safe action set at the equilibrium point of the system, e.g., $V_{\mathbb{A}_{\varphi},\text{episode}}/V_{\mathbb{A}_{\varphi},\text{equilibrium}}$. Figure 3b shows that action replacement relies significantly less on the safety mechanism than projection and masking. Generally, we report that a lower intervention rate often coincides with a higher reward.

**Comparison of learning tuples** We evaluate the impact of different learning tuples on the performance and intervention rates averaged over all five RL algorithms in Figure 4. When action masking is used, only safe actions can be sampled, i.e., only the *naive* tuple is meaningful; so we omit action masking from this evaluation. For both action replacement and projection, the *adaption penalty* tuple leads to the highest performance and lowest safety intervention rate, even outperforming the average over the baselines. In action projection, the *naive* tuple performs significantly worse than in action replacement. The *safe action* and *both* tuples seem to be only beneficial when using action projection and decrease performance when using action replacement.

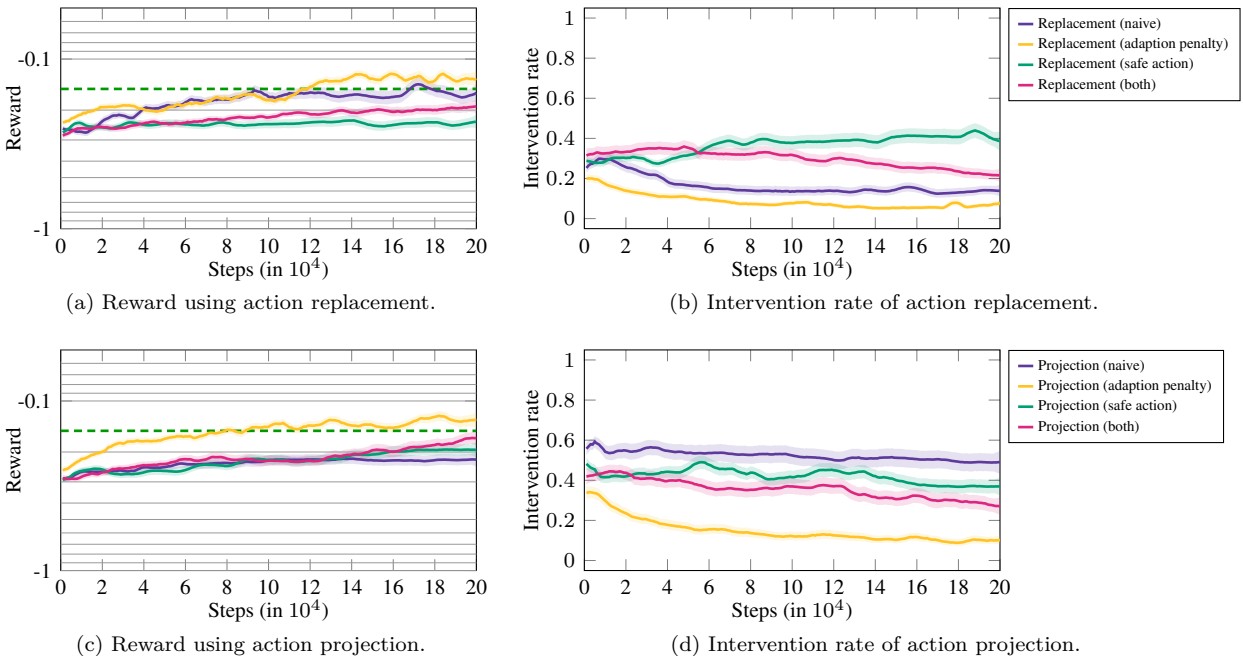

Figure 4: Evaluation of the training tuples for the 2D quadrotor averaged over the five RL algorithms TD3, SAC, DQN, continuous and discrete PPO on ten random seeds per algorithm. The left column depicts the reward and the right column the safety intervention rate. The top row shows the different learning tuples for action replacement and the bottom row for action projection.

## 5 Discussion

Our experiments confirm the theoretical statement that provably safe RL methods are always safe when Proposition 1 holds. In the two tested environments, the investigated RL baselines show non-zero safety violations during training, even after convergence. Subsequently, we discuss five statements resulting from the experiments, provide intuitions for implementing provably safe RL, summarize the limitations, and identify future research directions in provably safe RL.

**Selecting a provably safe RL approach** First, we want to summarize our experience with the three provably safe RL classes on the two investigated benchmarks. Action replacement was the easiest method to implement for continuous action spaces. It shows very good performance and low intervention rates. Hereby, using a failsafe action for replacement with an adaption penalty is simple to implement and was among the best-performing methods in our experiments. Still, the random sampling of safe actions might outperform the failsafe action. So, if sampling from the safe action space is readily available, e.g., in discrete action spaces, a failsafe controller is unnecessary. Action projection tends to be problematic in practice due to small numerical errors, resulting in infeasible optimization problems. We, therefore, have to reuse previous optimization results, e.g., as in Schürmann et al. (2018) for robust MPC, or use a failsafe controller if the optimization problem is not solvable. Together with the higher intervention rates compared to action replacement and the complex implementation, we would not recommend action projection based on our experience. However, if one already has a CBF or MPC formulation, it might be the most suitable solution. Action masking is particularly easy to implement for discrete action spaces and performs well in that setting. However, for action masking in continuous action spaces, there is no efficient and general algorithm yet that can handle safe action spaces significantly diverging from an axis-aligned box. Generally, the different approaches can also be combined to some extent. For example, if the optimization problem for action projection becomes infeasible, a failsafe controller can be used.

**Selecting a learning tuple** The *adaption penalty* learning tuple performed best, especially when using action projection. In our experiments, a simple constant reward penalty already improved the performance. Other environments may require more careful reward tuning, or the *adaption penalty* tuple could fail altogether due to reward hacking (Skalse et al., 2022) or goal misgeneralization (Langosco et al., 2022). The results in Figure 4 further show that using the safe action $\tilde{a}$ in the training tuple, i.e., configurations *both* and *safe action*, benefits the performance of action projection methods but impairs the training with action replacement. This effect can result from the fact that action replacement alters the action more than action projection, leading to a lower likelihood that the altered action stems from the RL policy. The evaluations in the Appendix show that this effect is prominent when using PPO. This on-policy algorithm assumes that the current batch of training data stems from the current policy. Hence, we would recommend using the *adaption penalty* tuple when possible and only using *safe action* in combination with off-policy methods.

**Convergence** The training of provably safe RL agents converges similarly fast or faster than the baselines in our experiments. In contrast, the performance (measured by the reward) at the beginning of the training is better for provably safe RL agents. One reason for the faster convergence is the exploration setting. Since $\mathbb{A}_\varphi \subseteq \mathbb{A}$, the provably safe agents learn in a usually smaller action space than the baselines, which can accelerate training. However, in some cases, the baseline agents might be better informed about the environment dynamics by exploring unsafe actions. Generally, the verification method should aim for $\mathbb{A}_\varphi = \mathbb{A}_s$ such that the provably safe agents can explore the full safe action space. Another reason for convergence differences can be that action replacement and action projection potentially correct the action after the forward pass through the policy. Thus, the gradient calculation might need correction as well. So far, there is only little theoretical work on this (Hunt et al., 2021; Gros et al., 2020), and it is unclear if, in practice, a correction of the gradient is necessary or if using the *adaption penalty* tuple is sufficient. Furthermore, the change in the distribution of the actions can impact the exploration strategy, as discussed in Section 2.4.

**Computational complexity** The computational complexity of the three approaches highly depends on the scenario-specific implementation. For action projection, the main implementation challenge is to guarantee that the optimization problem is always feasible. If the optimization problem can be formulated as a quadratic program, the computational complexity is polynomial, as shown by Vavasis (2001). On the contrary, the computational complexity of action replacement and action masking depends highly on the algorithm that identifies the safety of actions. For discrete action masking, the computational complexity apparently scales linearly with the total number of actions $\mathcal{O}(|\mathbb{A}|)$. For action replacement, we only need a single safe action, so in the ideal case, e.g., using a failsafe controller, the computational complexity is constant with respect to the total number of actions. Suppose an action replacement approach needs to determine the entire set of safe actions, it obviously has the same computational complexity as action masking with respect to the number of actions. The computational complexity for identifying the continuous safe action space depends on the task-specific implementation. One possibility is to compute the safe action space using set-based reachability analysis, where we want to point the interested reader to Althoff et al. (2021) for different approaches.

**Online vs. offline implementation** *Online* and *offline* have two notions in provably safe RL: online vs. offline safety verification and online vs. offline RL. The safety function usually needs to be evaluated online since, for continuous state spaces, pre-computing the safe action set for all states is often not feasible. Thus, the computational complexity of the safety function is important for real-time applications, as discussed in the previous paragraph. If the state and action space are discrete, it can be possible to pre-compute the safe actions offline (Alshiekh et al., 2018; Huang & Ontañón, 2022). Generally, safety is only guaranteed if the safety function is integrated between the agent and environment to correct actions (see Figure 1). In this study, we compare online on-policy and off-policy RL algorithms (Levine et al., 2020) since they are used for existing provably safe RL research. Still, provably safe RL can also be used for offline RL where the safety function would be integrated during deployment and most likely also during the data gathering phase if this phase is conducted in a safety-critical environment. However, more specific advice on offline provably safe RL needs to be substantiated with experimental evaluations and, thus, is a topic for future research.

**Limitations of provably safe RL**  There are limitations of provably safe RL that follow from the conceptual analysis in Section 2. Most importantly, safety has to be verifiable, i.e., there must be a safety function $\varphi(\boldsymbol{s}, \boldsymbol{a})$, which complies with Proposition 1. For this safety function, system knowledge is necessary, and especially for systems with a high number of continuous state variables, the safety function is potentially complex to compute. Additionally, safety guarantees are strongly tied to the safety function. If the safety function provides complex guarantees (e.g., ensures temporal logic specifications), it is usually computationally more expensive than for simpler guarantees (e.g., system stays within safe state set). Second, safety can only be decided if the state of the system is correctly observed within noise bounds. Thus, for a provably safe autonomous system, the perception module also needs to be verified such that it provides observations that are correct within the noise bounds. Third, there is often a trade-off between safety and performance since, for many tasks, these two objectives are only partially aligned. For example, if an automated vehicle drives faster, it reaches its destination earlier, but collisions are more difficult to avoid at higher speeds. Since provably safe RL ensures safe behavior, there is no such trade-off as safety is always prioritized over performance. Thus, the safety function $\varphi(\boldsymbol{s}, \boldsymbol{a})$ should not be too conservative since the agent would only perform trivial safe actions e.g., standing still at the side of the road forever. Lastly, comparing provably safe RL approaches is challenging as we need to define a safety function $\varphi(\boldsymbol{s}, \boldsymbol{a})$ that is efficiently usable by different approaches. Furthermore, the notion of safety is usually application-specific, so different application-specific approaches are hard to compare. We provide the first comparison of provably safe RL on two common continuous stabilization tasks, but further research is necessary to make more substantial claims about the most promising provably safe RL approaches.

**Future research based on proposed taxonomy**  Most action projection approaches discussed in Section 3 project the RL action on the border of $\mathbb{A}_\varphi$. In our experiments, we encountered two negative side effects related to this action projection implementation: First, the projection to the border of $\mathbb{A}_\varphi$ often leads to a relatively small $\mathbb{A}_\varphi$ in the next RL step, quickly resulting in a very small set $\mathbb{A}_\varphi$ if the RL agent proposes a few unsafe actions after each other. Second, small numerical errors can cause unsafe actions and must be considered in the safety verification. Therefore, future action projection research should investigate objective functions for (3) that achieve a more robust behavior while still depending on the action the RL agent proposed, e.g., projecting the action not to the border but by a learnable margin inside the provably safe action set. Action masking is a promising technique but has mainly been used with discrete action spaces in grid world environments and games, e.g., the Atari benchmark (Huang & Ontañón, 2022). Our proposed continuous action masking approach only applies to specific environments and performed well for the pendulum but showed mixed results for the 2D quadrotor. Thus, future research should investigate ways to extend continuous action masking to general convex or non-convex representations of $\mathbb{A}_\varphi$ to improve its applicability to more complex benchmarks and reduce the conservativeness of $\mathbb{A}_\varphi$. Additionally, it should be investigated if the agent should be informed about the reduction of the action space in action masking. This could result in improved convergence and an agent that is more aware of the action mask, similar to the effect of the *adaption penalty* tuple for action replacement and action projection. The evaluation of the considered benchmarks shows that action replacement performs better than action projection and masking, as discussed previously. However, it is still unclear how important the replacement strategy $\boldsymbol{\psi}(\boldsymbol{s})$ is for the convergence and performance of the agent, especially when applied to more complex tasks. Thus, future action replacement research should empirically and theoretically investigate this question.

**Improving the applicability of provably safe RL**  Despite the promising previous work discussed in Section 3, there are only few works on high-dimensional nonlinear systems and limited real-world applications. We suggest five major factors where future research would improve applicability. First, some approaches need to be computationally more efficient to be real-world applicable. The computational efficiency of verification methods is especially relevant and should be improved, as discussed previously. Second, we observe that the learning tuple used has a significant influence on the performance of the agent for some RL algorithms. Also, there needs to be more theoretical research on how provably safe RL approaches influence convergence to an optimal policy. More empirical and theoretical research on the effects of provably safe RL and its learning tuples for convergence is desirable. Third, common benchmarks are necessary to evaluate new provably safe RL approaches. Additionally, the three action correction strategies should be compared on more complex benchmarks to clarify if our observations can be extended to them. Such benchmarking

would make research on provably safe RL more comparable, ease starting research on provably safe RL, and provide more evidence to decide on the best-suited provably safe RL approach. Fourth, recent work shows a low variety of safety specifications, mainly comprising stabilization and reach-avoid specifications. On the contrary, real-world safety is more complex, e.g., traffic rules such as waiting at a red light and safely but quickly moving at a green light. Finally, provably safe RL requires expert knowledge of verification methods. Future research could mitigate this through modular and automatic approaches, where fewer engineering decisions are necessary and more parameters are tuned automatically. With these advances, provably safe RL could bring the best elements of RL and formal specifications together towards RL methods that require as little expert knowledge as necessary and provide formal guarantees for complex safety specifications to achieve reliable and trustworthy cyber-physical systems.

## 6 Conclusion

In conclusion, we categorize provably safe RL methods to structure the literature from a machine learning perspective. We present provably safe RL methods from a conceptual perspective and discuss necessary assumptions. Our proposed categorization into action replacement, action projection, and action masking supports researchers in comparing their works and provides valuable insights into the selection process of provably safe RL methods. The comparison of four implementations of provably safe RL on a 2D quadrotor and an inverted pendulum stabilization benchmark provides further insights into the best-suited method for different tasks. We further present practical recommendations for selecting a provably safe RL approach and a learning tuple, which will be valuable for researchers who are new to RL or formal methods. Lastly, as discussed in Section 5, our proposed taxonomy and experimental evaluation yield multiple promising future research directions.

### Acknowledgments

The authors gratefully acknowledge the partial financial support of this work by the research training group ConVeY, funded by the German Research Foundation under grant GRK 2428, by the project TRAITS under grant number 01IS21087, funded by the German Federal Ministry of Education and Research, by the Horizon 2020 EU Framework Project CONCERT under grant number 101016007, by the project justITSELF funded by the European Research Council (ERC) under grant agreement number 817629, and by the German Federal Ministry for Economics Affairs and Climate Action project VaF under grant number KK5135901KG0.

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

## A   Appendix

**MDP modification with action replacement**

Action replacement alters the MDP on which the agent learns. Hunt et al. (2021) discuss this modification for discrete action spaces and uniformly sampling from the safe action space. We generalize this discussion to using any replacement function and continuous action spaces. To this end, we define $\boldsymbol{\psi}(\boldsymbol{s})$ so that it randomly samples the replacement action $\tilde{\boldsymbol{a}}$ according to a replacement policy $\boldsymbol{\pi}_r\left(\tilde{\boldsymbol{a}}|\boldsymbol{s}\right)$ with $\sum_{\tilde{\boldsymbol{a}}\in\mathbb{A}_\varphi(\boldsymbol{s})}\boldsymbol{\pi}_r\left(\tilde{\boldsymbol{a}}|\boldsymbol{s}\right)=1$ for the discrete case, and $\int_{\mathbb{A}_\varphi(\boldsymbol{s})}\boldsymbol{\pi}_r\left(\tilde{\boldsymbol{a}}|\boldsymbol{s}\right)d\tilde{\boldsymbol{a}}=1$ for the continuous case, and $\forall\tilde{\boldsymbol{a}}\in\mathbb{A}_\varphi(\boldsymbol{s}):\boldsymbol{\pi}_r\left(\tilde{\boldsymbol{a}}|\boldsymbol{s}\right)\geq 0$. In the example of uniform sampling from $\mathbb{A}_\varphi(\boldsymbol{s})$, the replacement policy is $\boldsymbol{\pi}_r\left(\tilde{\boldsymbol{a}}|\boldsymbol{s}\right)=1/V_{\mathbb{A}_\varphi(\boldsymbol{s})}$ where $V_{\mathbb{A}_\varphi(\boldsymbol{s})}$ is the volume of $\mathbb{A}_\varphi(\boldsymbol{s})$. By replacing unsafe actions, the transition function of the MDP changes to

$$T_\varphi(\boldsymbol{s},\boldsymbol{a},\boldsymbol{s}')=\begin{cases}T(\boldsymbol{s},\boldsymbol{a},\boldsymbol{s}'), & \text{if } \varphi(\boldsymbol{s},\boldsymbol{a})=1\\T_r(\boldsymbol{s},\boldsymbol{s}'), & \text{otherwise,}\end{cases} \tag{15}$$

$$T_r(\boldsymbol{s},\boldsymbol{s}')=\sum_{\tilde{\boldsymbol{a}}\in\mathbb{A}_\varphi(\boldsymbol{s})}\boldsymbol{\pi}_r\left(\tilde{\boldsymbol{a}}|\boldsymbol{s}\right)T(\boldsymbol{s},\tilde{\boldsymbol{a}},\boldsymbol{s}'). \tag{16}$$

The reward function of the MDP changes accordingly to

$$r_\varphi(\boldsymbol{s},\boldsymbol{a})=\begin{cases}r(\boldsymbol{s},\boldsymbol{a}), & \text{if } \varphi(\boldsymbol{s},\boldsymbol{a})=1\\r_r(\boldsymbol{s}), & \text{otherwise,}\end{cases} \tag{17}$$

$$r_r(\boldsymbol{s})=\sum_{\tilde{\boldsymbol{a}}\in\mathbb{A}_\varphi(\boldsymbol{s})}\boldsymbol{\pi}_r\left(\tilde{\boldsymbol{a}}|\boldsymbol{s}\right)r(\boldsymbol{s},\tilde{\boldsymbol{a}}). \tag{18}$$

In the continuous case, we get $T_r(\boldsymbol{s},\boldsymbol{s}')$ by marginalizing the transition probability density function over $\mathbb{A}_\varphi(\boldsymbol{s})$:

$$T_r(\boldsymbol{s},\boldsymbol{s}')=\int_{\mathbb{A}_\varphi(\boldsymbol{s})}\boldsymbol{\pi}_r\left(\tilde{\boldsymbol{a}}|\boldsymbol{s}\right)T(\boldsymbol{s},\tilde{\boldsymbol{a}},\boldsymbol{s}')d\tilde{\boldsymbol{a}}. \tag{19}$$

Analogously, we have that

$$r_r(\boldsymbol{s})=\int_{\mathbb{A}_\varphi(\boldsymbol{s})}\boldsymbol{\pi}_r\left(\tilde{\boldsymbol{a}}|\boldsymbol{s}\right)r(\boldsymbol{s},\tilde{\boldsymbol{a}})d\tilde{\boldsymbol{a}}. \tag{20}$$

## Environment parameters

We provide an overview of all environment-specific parameters in Table 3 and Table 4.

Table 3: Environment parameters of the pendulum.

| Parameter | Value |
|-----------|-------|
| Gravity $g$ | $9.81\,\mathrm{m\,s^{-2}}$ |
| Mass $m$ | $1\,\mathrm{kg}$ |
| Length $l$ | $1\,\mathrm{m}$ |

## Hyperparameters for learning algorithms

We specify the hyperparameters for all learning algorithms (see Table 5 for PPO, Table 6 for TD3, Table 7 for DQN, and Table 8 for SAC) that are different from the Stable Baselines3 (Raffin et al., 2021) default values. Additionally, the code for the experiments is available at the CodeOcean capsule doi.org/10.24433/CO.9209121.v1  to reproduce our results.

Table 4: Environment parameters of the 2D quadrotor.

| Parameter | Value |
|---|---|
| Gravity $g$ | $9.81\,\mathrm{m\,s^{-2}}$ |
| $k$ | $1\,1/\mathrm{kg}$ |
| $d_0$ | 70 |
| $d_1$ | 17 |
| $n_0$ | 55 |
| $\mathbb{W}$ | $[[-0.1, 0.1], [-0.1, 0.1]]$ |

Table 5: Hyperparameters for PPO.

| Parameter | Pendulum | 2D quadrotor |
|---|---|---|
| Learning rate | $1 \times 10^{-4}$ | $5 \times 10^{-5}$ |
| Discount factor $\gamma$ | 0.98 | 0.999 |
| Steps per update | 2048 | 512 |
| Optimization epochs | 20 | 30 |
| Minibatch size | 16 | 128 |
| Max gradient clipping | 0.9 | 0.5 |
| Entropy coefficient | $1 \times 10^{-3}$ | $2 \times 10^{-6}$ |
| Value function coefficient | 0.045 | 0.5 |
| Clipping range | 0.3 | 0.1 |
| Generalized advantage estimation $\lambda$ | 0.8 | 0.92 |
| Activation function | ReLU | ReLU |
| Hidden layers | 2 | 2 |
| Neurons per layer | 32 | 64 |
| Training steps | 60k | 200k |

Table 6: Hyperparameters for TD3.

| Parameter | Pendulum | 2D quadrotor |
|---|---|---|
| Learning rate | $3.5 \times 10^{-3}$ | $2 \times 10^{-3}$ |
| Replay buffer size | $1 \times 10^{4}$ | $1 \times 10^{5}$ |
| Discount factor $\gamma$ | 0.98 | 0.98 |
| Initial exploration steps | $10 \times 10^{3}$ | 100 |
| Steps between model updates | 256 | 5 |
| Gradient steps per model update | 256 | 10 |
| Minibatch size per gradient step | 512 | 512 |
| Soft update coefficient $\tau$ | $5 \times 10^{-3}$ | $5 \times 10^{-3}$ |
| Gaussian smoothing noise $\sigma$ | 0.2 | 0.12 |
| Activation function | ReLU | ReLU |
| Hidden layers | 2 | 2 |
| Neurons per layer | 32 | 64 |
| Training steps | 60k | 200k |

Table 7: Hyperparameters for DQN.

| Parameter | Pendulum | 2D quadrotor |
|---|---|---|
| Learning rate | $2 \times 10^{-3}$ | $1 \times 10^{-4}$ |
| Replay buffer size | $5 \times 10^{4}$ | $1 \times 10^{6}$ |
| Discount factor $\gamma$ | 0.95 | 0.999 99 |
| Initial exploration steps | 500 | 100 |
| Steps between model updates | 8 | 2 |
| Gradient steps per model update | 4 | 4 |
| Minibatch size per gradient step | 512 | 64 |
| Maximum for gradient clipping | 10 | 100 |
| Update frequency target network | $1 \times 10^{3}$ | $1 \times 10^{3}$ |
| Initial exploration probability $\epsilon$ | 1.0 | 0.137 |
| Linear interpolation steps of $\epsilon$ | $6 \times 10^{3}$ | $1 \times 10^{4}$ |
| Final exploration probability $\epsilon$ | 0.1 | 0.004 |
| Activation function | tanh | tanh |
| Hidden layers | 2 | 2 |
| Neurons per layer | 32 | 64 |
| Training steps | 60k | 200k |

Table 8: Hyperparameters for SAC.

| Parameter | Pendulum | 2D quadrotor |
|---|---|---|
| Learning rate | $3 \times 10^{-4}$ | $3 \times 10^{-4}$ |
| Replay buffer size | $1 \times 10^{6}$ | $5 \times 10^{5}$ |
| Discount factor $\gamma$ | 0.99 | 0.98 |
| Initial exploration steps | 100 | 1000 |
| Steps between model updates | 1 | 32 |
| Gradient steps per model update | 1 | 32 |
| Minibatch size per gradient step | 256 | 512 |
| Entropy coefficient | learned | $1 \times 10^{-1}$ |
| Soft update coefficient $\tau$ | $5 \times 10^{-3}$ | $1 \times 10^{-2}$ |
| Activation function | ReLU | ReLU |
| Hidden layers | 2 | 2 |
| Neurons per layer | 32 | 64 |
| Training steps | 60k | 200k |

## Full evaluation

In this section, we present all training results of the five RL algorithms TD3, SAC, DQN, and PPO continuous and discrete. We compare these algorithms on the inverted pendulum and 2D quadrotor environment on ten random seeds. The tested algorithms are action replacement with $\boldsymbol{\psi}_{\mathrm{sample}}(\boldsymbol{s})$ and $\boldsymbol{\psi}_{\mathrm{failsafe}}(\boldsymbol{s})$, action projection, and action masking. When action replacement is used with a failsafe controller, we omit *safe action* and *both* because for discrete action spaces, the failsafe controller might use an action that is not in the discrete action space. This is due to the failsafe controller proposing actions from the continuous action space.

First, we present the effect of the learning tuples on the on-policy algorithm PPO in Figure 5. This comparison clearly shows the negative effect of the *safe action* tuple on the training performance of PPO. Figure 6 depicts the aggregated training results for the pendulum as previously discussed for the 2D quadrotor in Figures 3, 4 and 5. Figures 7 to 10 depict how the reward and intervention rate evolve during training for all investigated configurations. Finally, the Tables 9 and 10 show statistical results for deploying the learned models for the two benchmarks.

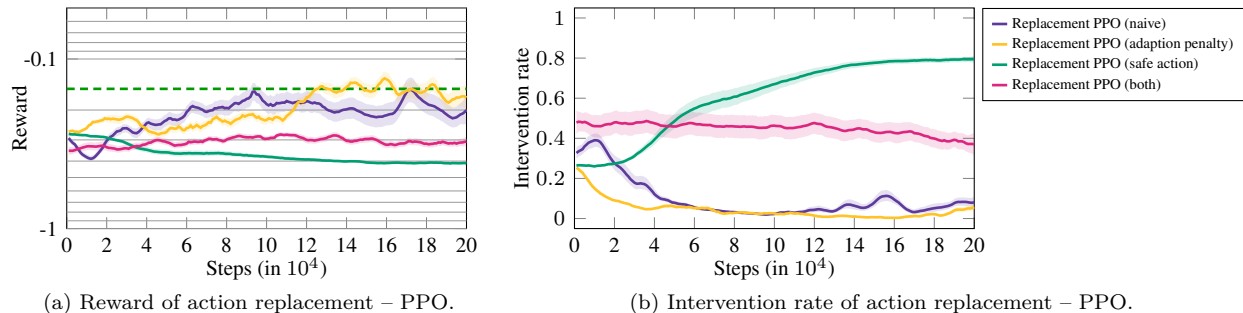

(a) Reward of action replacement – PPO.  (b) Intervention rate of action replacement – PPO.

Figure 5: Evaluation of the training tuples for the 2D quadrotor averaged over the continuous and discrete PPO training runs using action replacement with ten random seeds each. The left column depicts the reward and the right column the safety intervention rate.

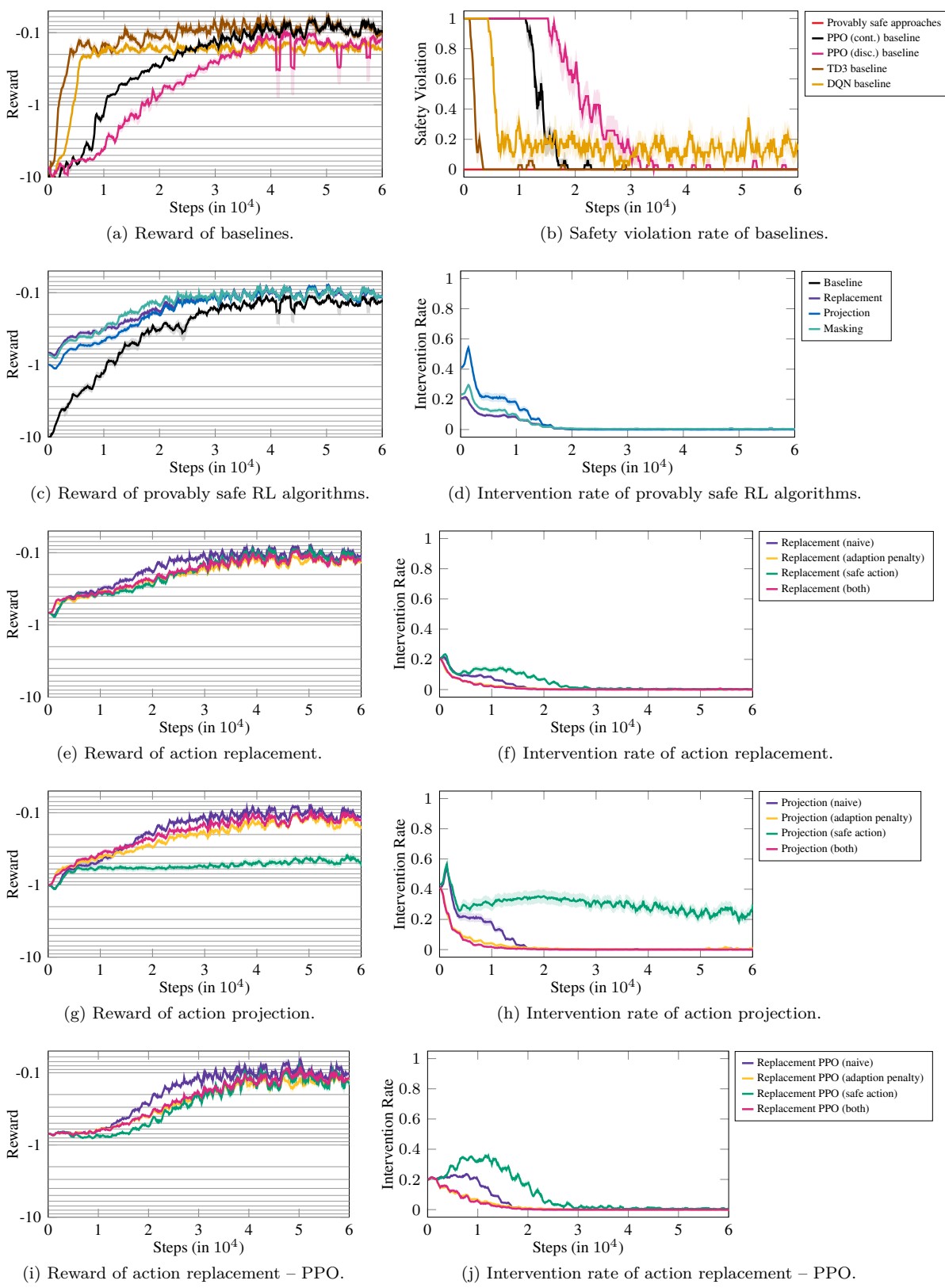

Figure 6: Evaluation of the training tuples for the pendulum averaged over ten random seeds each. The left column depicts the reward and the right column the safety intervention rate. We would like to refer the reader to Figures 3, 4 and 5 for the corresponding 2D quadrotor results.

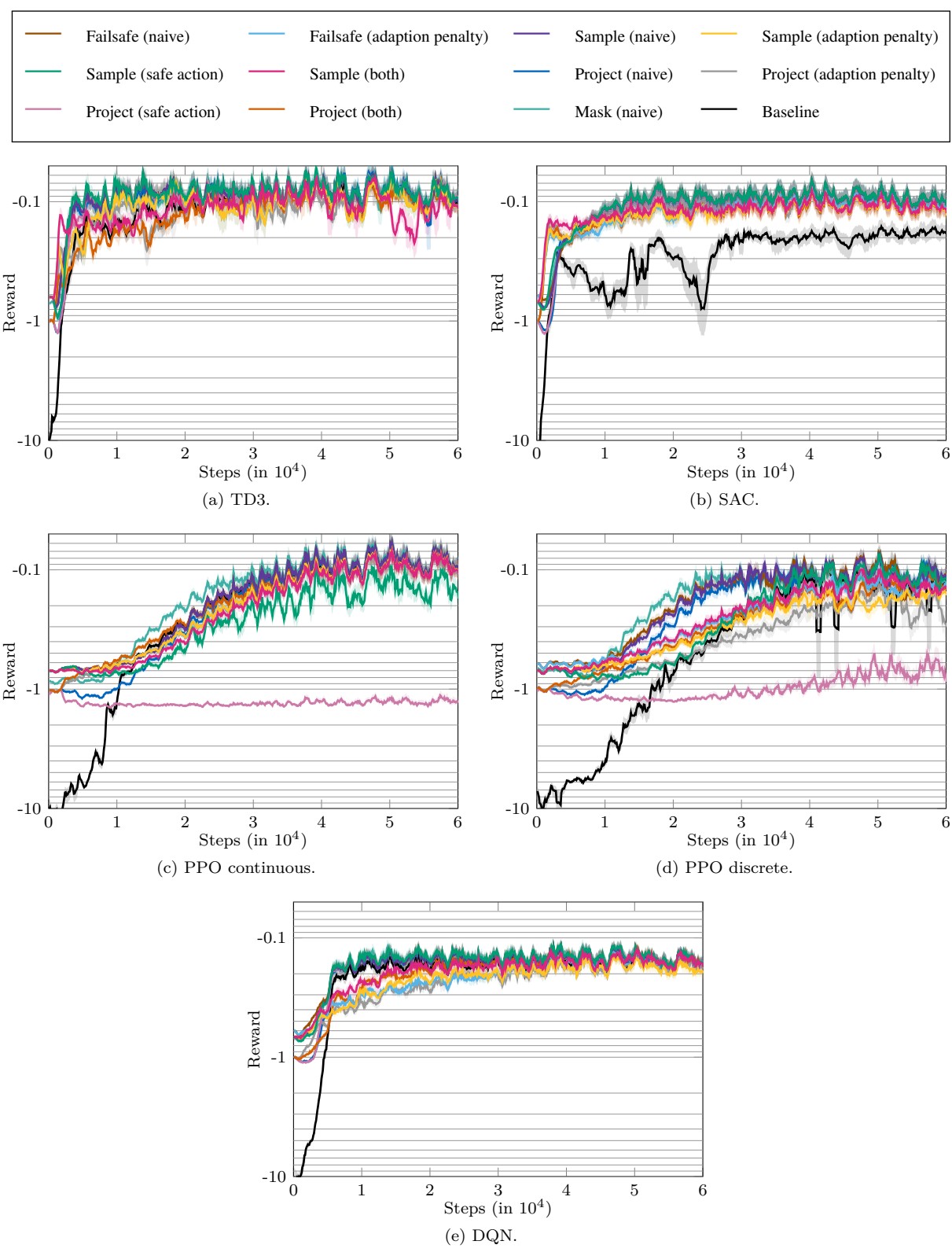

Figure 7: **Pendulum:** Average reward and standard deviation per training step for TD3, SAC, DQN, PPO discrete, and PPO continuous. For each configuration, ten training runs with different random seeds were conducted. Each subplot contains all implemented variants. Note that the reward for the *adaption penalty* variants is still $r$ and the adaption penalty $r^*$ is not included in the curves for better comparability.

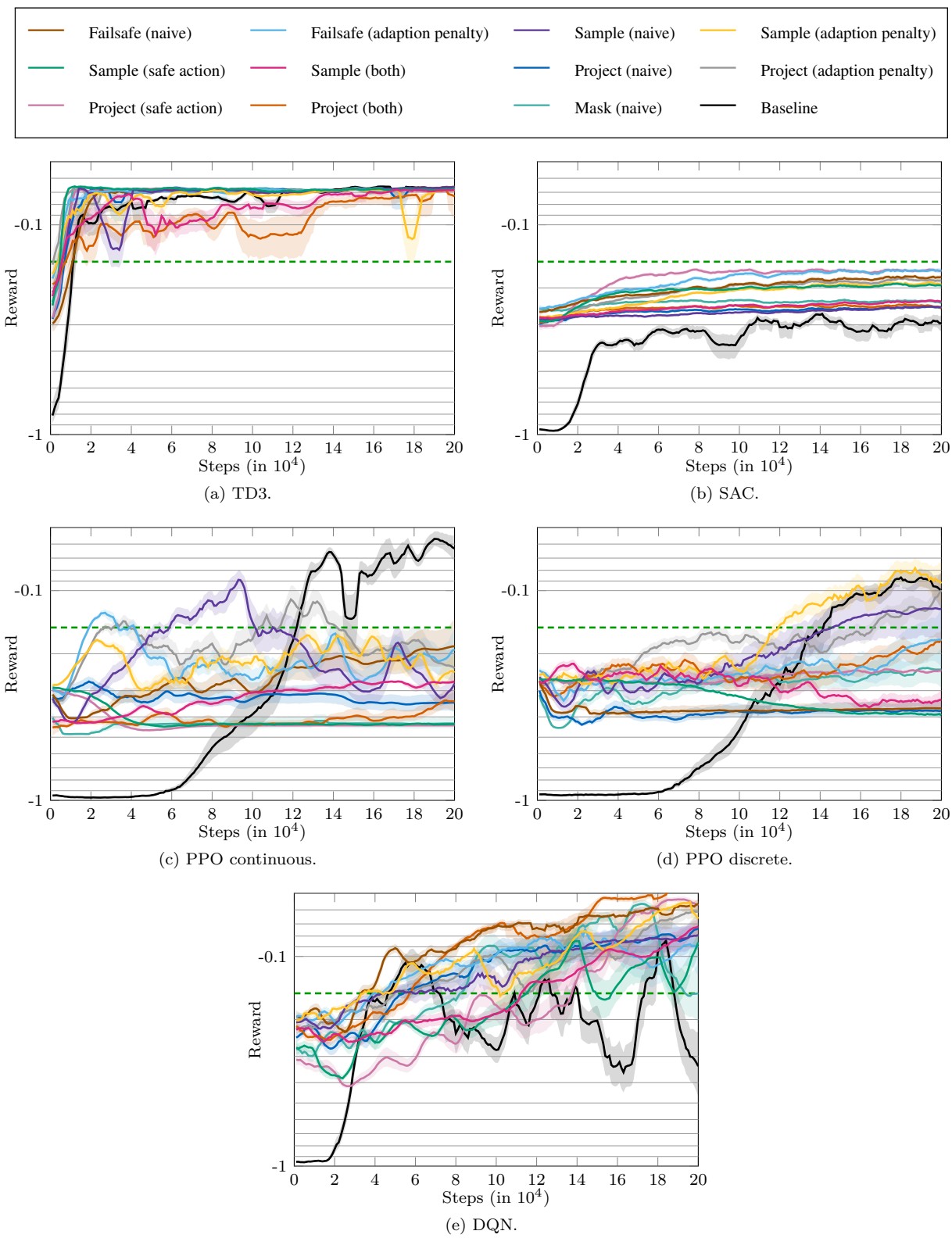

Figure 8: **2D quadrotor:** Average reward and standard deviation per training step for TD3, SAC, DQN, PPO discrete, and PPO continuous. For each configuration, ten training runs with different random seeds were conducted. Each subplot contains all implemented variants. Note that the reward for the *adaption penalty* variants is still $r$ and the adaption penalty $r^*$ is not included in the curves for better comparability.

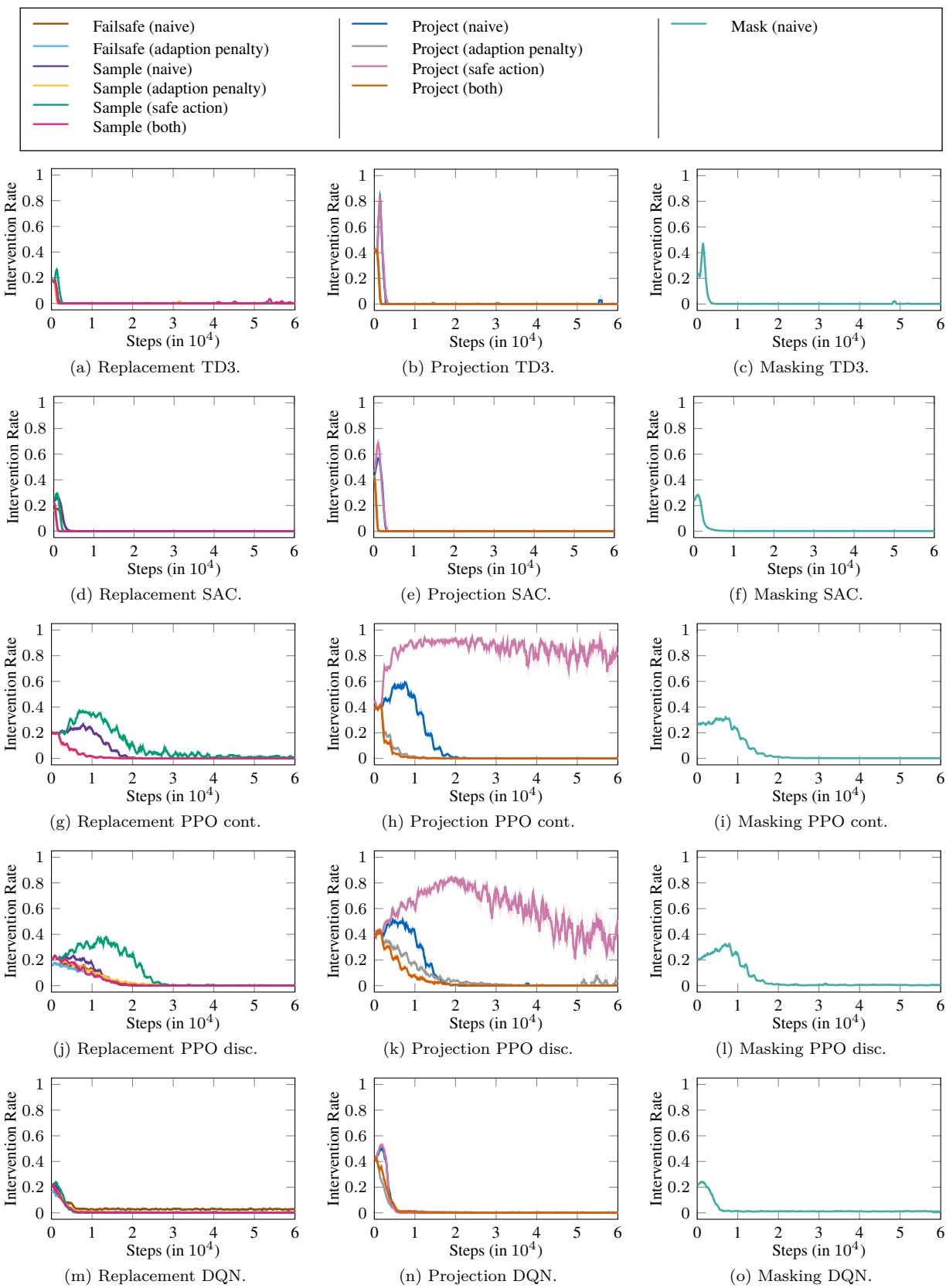

Figure 9: **Pendulum:** Intervention rate for TD3, SAC, PPO discrete, PPO continuous, and DQN.

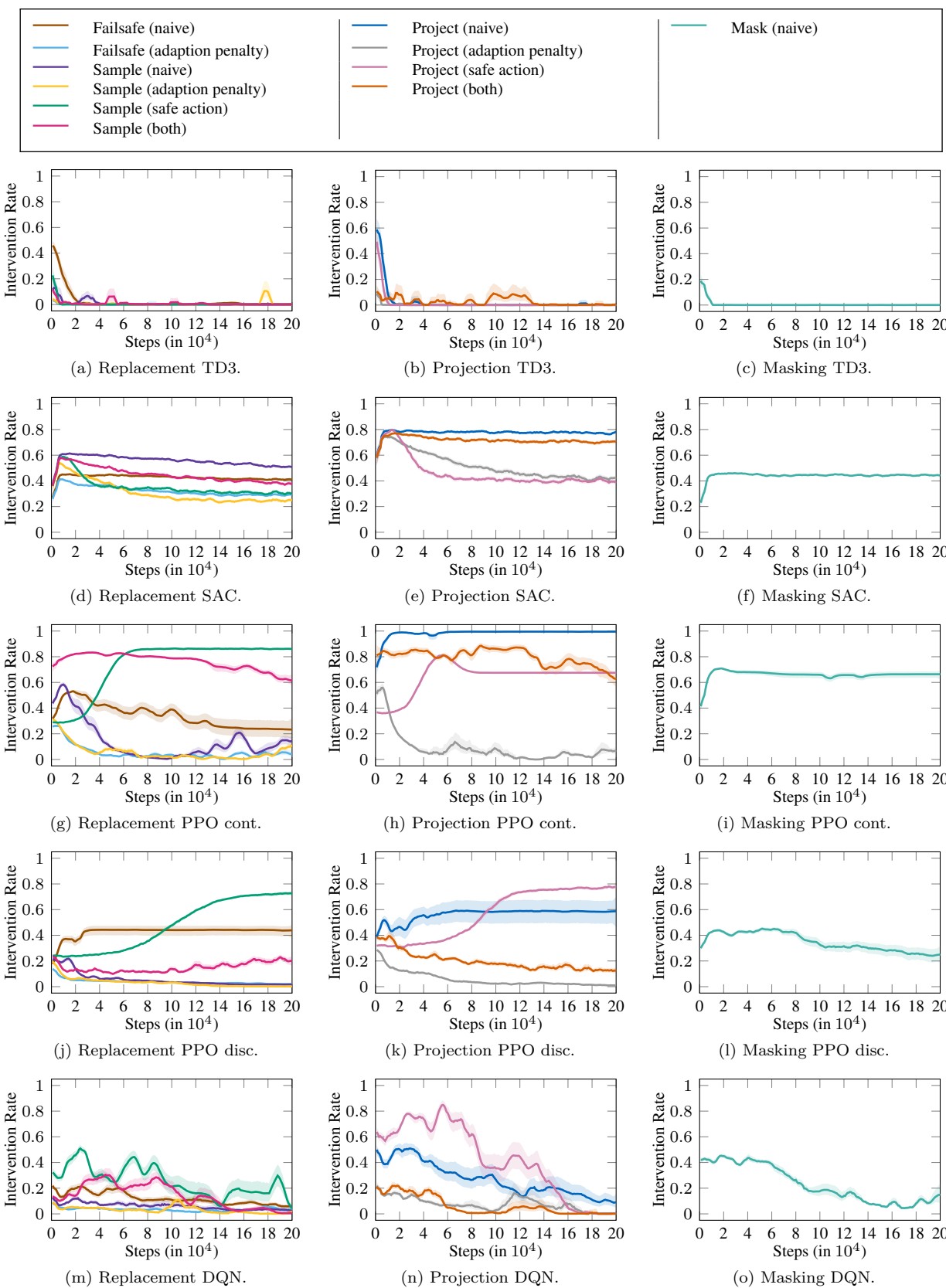

Figure 10: **2D quadrotor:** Intervention rate for TD3, SAC, PPO discrete, PPO continuous. and DQN.

Table 9: Mean and standard deviation of 30 pendulum deployment episodes.

| Approach | Reward | | Intervention Rate | | Safety Violation | |
|---|---|---|---|---|---|---|
| | MEAN | STD. DEV. | MEAN | STD. DEV. | MEAN | STD. DEV. |
| **PPO (continuous)** | | | | | | |
| PROJECTION (SAFEACTION) | -1.14 | 0.42 | 0.76 | 0.29 | 0.00 | 0.00 |
| PROJECTION (ADAPTIONPENALTY) | -0.07 | 0.07 | 0.00 | 0.00 | 0.00 | 0.00 |
| PROJECTION (BOTH) | -0.07 | 0.07 | 0.00 | 0.00 | 0.00 | 0.00 |
| PROJECTION (NAIVE) | -0.06 | 0.07 | 0.00 | 0.00 | 0.00 | 0.00 |
| SAMPLE (SAFEACTION) | -0.13 | 0.16 | 0.01 | 0.02 | 0.00 | 0.00 |
| SAMPLE (ADAPTIONPENALTY) | -0.07 | 0.07 | 0.00 | 0.00 | 0.00 | 0.00 |
| SAMPLE (BOTH) | -0.07 | 0.07 | 0.00 | 0.00 | 0.00 | 0.00 |
| SAMPLE (NAIVE) | -0.07 | 0.07 | 0.00 | 0.00 | 0.00 | 0.00 |
| FAILSAFE (ADAPTIONPENALTY) | -0.07 | 0.07 | 0.00 | 0.00 | 0.00 | 0.00 |
| FAILSAFE (NAIVE) | -0.07 | 0.07 | 0.00 | 0.00 | 0.00 | 0.00 |
| BASELINE (NAIVE) | -0.06 | 0.07 | — | — | 0.00 | 0.00 |
| MASKING (NAIVE) | -0.07 | 0.07 | 0.00 | 0.00 | 0.00 | 0.00 |
| **PPO (discrete)** | | | | | | |
| PROJECTION (SAFEACTION) | -0.52 | 0.55 | 0.28 | 0.42 | 0.00 | 0.00 |
| PROJECTION (ADAPTIONPENALTY) | -0.14 | 0.12 | 0.00 | 0.00 | 0.00 | 0.00 |
| PROJECTION (BOTH) | -0.09 | 0.09 | 0.00 | 0.00 | 0.00 | 0.00 |
| PROJECTION (NAIVE) | -0.15 | 0.27 | 0.03 | 0.10 | 0.00 | 0.00 |
| SAMPLE (SAFEACTION) | -0.09 | 0.08 | 0.00 | 0.00 | 0.00 | 0.00 |
| SAMPLE (ADAPTIONPENALTY) | -0.13 | 0.16 | 0.00 | 0.00 | 0.00 | 0.00 |
| SAMPLE (BOTH) | -0.10 | 0.08 | 0.00 | 0.00 | 0.00 | 0.00 |
| SAMPLE (NAIVE) | -0.09 | 0.08 | 0.00 | 0.00 | 0.00 | 0.00 |
| FAILSAFE (ADAPTIONPENALTY) | -0.09 | 0.07 | 0.00 | 0.00 | 0.00 | 0.00 |
| FAILSAFE (NAIVE) | -0.08 | 0.07 | 0.00 | 0.00 | 0.00 | 0.00 |
| BASELINE (NAIVE) | -0.08 | 0.07 | — | — | 0.00 | 0.00 |
| MASKING (NAIVE) | -0.07 | 0.07 | 0.00 | 0.00 | 0.00 | 0.00 |
| **TD3** | | | | | | |
| PROJECTION (SAFEACTION) | -0.07 | 0.07 | 0.00 | 0.00 | 0.00 | 0.00 |
| PROJECTION (ADAPTIONPENALTY) | -0.08 | 0.08 | 0.00 | 0.00 | 0.00 | 0.00 |
| PROJECTION (BOTH) | -0.07 | 0.07 | 0.00 | 0.00 | 0.00 | 0.00 |
| PROJECTION (NAIVE) | -0.07 | 0.07 | 0.00 | 0.00 | 0.00 | 0.00 |
| SAMPLE (SAFEACTION) | -0.07 | 0.07 | 0.00 | 0.00 | 0.00 | 0.00 |
| SAMPLE (ADAPTIONPENALTY) | -0.09 | 0.07 | 0.00 | 0.00 | 0.00 | 0.00 |
| SAMPLE (BOTH) | -0.09 | 0.07 | 0.00 | 0.00 | 0.00 | 0.00 |
| SAMPLE (NAIVE) | -0.07 | 0.07 | 0.00 | 0.00 | 0.00 | 0.00 |
| FAILSAFE (ADAPTIONPENALTY) | -0.09 | 0.07 | 0.00 | 0.00 | 0.00 | 0.00 |
| FAILSAFE (NAIVE) | -0.07 | 0.07 | 0.00 | 0.00 | 0.00 | 0.00 |
| BASELINE (NAIVE) | -0.07 | 0.07 | — | — | 0.00 | 0.00 |
| MASKING (NAIVE) | -0.07 | 0.07 | 0.00 | 0.00 | 0.00 | 0.00 |
| **DQN** | | | | | | |
| PROJECTION (SAFEACTION) | -0.07 | 0.07 | 0.00 | 0.00 | 0.00 | 0.00 |
| PROJECTION (ADAPTIONPENALTY) | -0.07 | 0.07 | 0.00 | 0.00 | 0.00 | 0.00 |
| PROJECTION (BOTH) | -0.07 | 0.08 | 0.00 | 0.00 | 0.00 | 0.00 |
| PROJECTION (NAIVE) | -0.07 | 0.07 | 0.00 | 0.00 | 0.00 | 0.00 |
| SAMPLE (SAFEACTION) | -0.07 | 0.07 | 0.00 | 0.00 | 0.00 | 0.00 |
| SAMPLE (ADAPTIONPENALTY) | -0.09 | 0.07 | 0.00 | 0.00 | 0.00 | 0.00 |
| SAMPLE (BOTH) | -0.07 | 0.08 | 0.00 | 0.00 | 0.00 | 0.00 |
| SAMPLE (NAIVE) | -0.07 | 0.07 | 0.00 | 0.00 | 0.00 | 0.00 |
| FAILSAFE (ADAPTIONPENALTY) | -0.07 | 0.07 | 0.00 | 0.00 | 0.00 | 0.00 |
| FAILSAFE (NAIVE) | -0.07 | 0.07 | 0.00 | 0.00 | 0.00 | 0.00 |
| BASELINE (NAIVE) | -0.07 | 0.07 | — | — | 0.00 | 0.00 |
| MASKING (NAIVE) | -0.07 | 0.07 | 0.00 | 0.00 | 0.00 | 0.00 |
| **SAC** | | | | | | |
| PROJECTION (NAIVE) | -0.08 | 0.07 | 0.00 | 0.00 | 0.00 | 0.00 |
| PROJECTION (ADAPTIONPENALTY) | -0.10 | 0.09 | 0.00 | 0.00 | 0.00 | 0.00 |
| PROJECTION (SAFEACTION) | -0.08 | 0.07 | 0.00 | 0.00 | 0.00 | 0.00 |
| PROJECTION (BOTH) | -0.10 | 0.08 | 0.00 | 0.00 | 0.00 | 0.00 |
| SAMPLE (NAIVE) | -0.08 | 0.07 | 0.00 | 0.00 | 0.00 | 0.00 |
| SAMPLE (ADAPTIONPENALTY) | -0.10 | 0.08 | 0.00 | 0.00 | 0.00 | 0.00 |
| SAMPLE (SAFEACTION) | -0.08 | 0.07 | 0.00 | 0.00 | 0.00 | 0.00 |
| SAMPLE (BOTH) | -0.09 | 0.08 | 0.00 | 0.00 | 0.00 | 0.00 |
| FAILSAFE (NAIVE) | -0.08 | 0.07 | 0.00 | 0.00 | 0.00 | 0.00 |
| FAILSAFE (ADAPTIONPENALTY) | -0.09 | 0.09 | 0.00 | 0.00 | 0.00 | 0.00 |
| BASELINE (NAIVE) | -0.11 | 0.09 | — | — | 0.00 | 0.00 |
| MASKING (NAIVE) | -0.08 | 0.07 | 0.00 | 0.00 | 0.00 | 0.00 |

Note: — indicates that there is no intervention rate for the baselines as they don't implement a safety verification.

Table 10: Mean and standard deviation of 30 2D Quadrotor deployment episodes.

| Approach | Reward | | Intervention Rate | | Safety Violation | |
|---|---|---|---|---|---|---|
| | MEAN | STD. DEV. | MEAN | STD. DEV. | MEAN | STD. DEV. |
| **PPO (continuous)** | | | | | | |
| PROJECTION (SAFEACTION) | -0.44 | 0.00 | 0.68 | 0.00 | 0.00 | 0.00 |
| PROJECTION (ADAPTIONPENALTY) | -0.33 | 0.07 | 0.44 | 0.05 | 0.00 | 0.00 |
| PROJECTION (BOTH) | -0.39 | 0.07 | 0.57 | 0.13 | 0.00 | 0.00 |
| PROJECTION (NAIVE) | -0.31 | 0.10 | 0.47 | 0.25 | 0.00 | 0.00 |
| SAMPLE (SAFEACTION) | -0.43 | 0.00 | 0.86 | 0.00 | 0.00 | 0.00 |
| SAMPLE (ADAPTIONPENALTY) | -0.28 | 0.12 | 0.10 | 0.12 | 0.00 | 0.00 |
| SAMPLE (BOTH) | -0.36 | 0.03 | 0.41 | 0.20 | 0.00 | 0.00 |
| SAMPLE (NAIVE) | -0.39 | 0.06 | 0.23 | 0.13 | 0.00 | 0.00 |
| FAILSAFE (ADAPTIONPENALTY) | -0.31 | 0.11 | 0.08 | 0.06 | 0.00 | 0.00 |
| FAILSAFE (NAIVE) | -0.27 | 0.11 | 0.27 | 0.29 | 0.00 | 0.00 |
| BASELINE (NAIVE) | -0.86 | 0.01 | — | — | 0.94 | 0.01 |
| MASKING (NAIVE) | -0.43 | 0.09 | 0.57 | 0.28 | 0.00 | 0.00 |
| **PPO (discrete)** | | | | | | |
| PROJECTION (SAFEACTION) | -0.44 | 0.01 | 0.74 | 0.21 | 0.00 | 0.00 |
| PROJECTION (ADAPTIONPENALTY) | -0.24 | 0.13 | 0.02 | 0.02 | 0.00 | 0.00 |
| PROJECTION (BOTH) | -0.35 | 0.11 | 0.16 | 0.14 | 0.00 | 0.00 |
| PROJECTION (NAIVE) | -0.34 | 0.14 | 0.46 | 0.37 | 0.00 | 0.00 |
| SAMPLE (SAFEACTION) | -0.42 | 0.02 | 0.82 | 0.01 | 0.00 | 0.00 |
| SAMPLE (ADAPTIONPENALTY) | -0.11 | 0.10 | 0.00 | 0.00 | 0.00 | 0.00 |
| SAMPLE (BOTH) | -0.38 | 0.09 | 0.32 | 0.18 | 0.00 | 0.00 |
| SAMPLE (NAIVE) | -0.13 | 0.16 | 0.02 | 0.03 | 0.00 | 0.00 |
| FAILSAFE (ADAPTIONPENALTY) | -0.19 | 0.15 | 0.01 | 0.03 | 0.00 | 0.00 |
| FAILSAFE (NAIVE) | -0.34 | 0.13 | 0.41 | 0.21 | 0.00 | 0.00 |
| BASELINE (NAIVE) | -0.11 | 0.03 | — | — | 0.00 | 0.00 |
| MASKING (NAIVE) | -0.25 | 0.14 | 0.28 | 0.21 | 0.00 | 0.00 |
| **TD3** | | | | | | |
| PROJECTION (SAFEACTION) | -0.21 | 0.03 | 0.28 | 0.10 | 0.00 | 0.00 |
| PROJECTION (ADAPTIONPENALTY) | -0.22 | 0.03 | 0.26 | 0.10 | 0.00 | 0.00 |
| PROJECTION (BOTH) | -0.21 | 0.04 | 0.28 | 0.12 | 0.00 | 0.00 |
| PROJECTION (NAIVE) | -0.25 | 0.05 | 0.26 | 0.17 | 0.00 | 0.00 |
| SAMPLE (SAFEACTION) | -0.18 | 0.03 | 0.07 | 0.01 | 0.00 | 0.00 |
| SAMPLE (ADAPTIONPENALTY) | -0.21 | 0.04 | 0.13 | 0.05 | 0.00 | 0.00 |
| SAMPLE (BOTH) | -0.21 | 0.06 | 0.06 | 0.03 | 0.00 | 0.00 |
| SAMPLE (NAIVE) | -0.19 | 0.04 | 0.12 | 0.09 | 0.00 | 0.00 |
| FAILSAFE (ADAPTIONPENALTY) | -0.20 | 0.03 | 0.05 | 0.02 | 0.00 | 0.00 |
| FAILSAFE (NAIVE) | -0.26 | 0.09 | 0.05 | 0.02 | 0.00 | 0.00 |
| BASELINE (NAIVE) | -0.90 | 0.05 | — | — | 0.95 | 0.02 |
| MASKING (NAIVE) | -0.16 | 0.03 | 0.03 | 0.03 | 0.00 | 0.00 |
| **DQN** | | | | | | |
| PROJECTION (SAFEACTION) | -0.06 | 0.02 | 0.00 | 0.01 | 0.00 | 0.00 |
| PROJECTION (ADAPTIONPENALTY) | -0.05 | 0.00 | 0.00 | 0.00 | 0.00 | 0.00 |
| PROJECTION (BOTH) | -0.05 | 0.01 | 0.00 | 0.00 | 0.00 | 0.00 |
| PROJECTION (NAIVE) | -0.09 | 0.06 | 0.12 | 0.24 | 0.00 | 0.00 |
| SAMPLE (SAFEACTION) | -0.07 | 0.01 | 0.01 | 0.02 | 0.00 | 0.00 |
| SAMPLE (ADAPTIONPENALTY) | -0.06 | 0.03 | 0.00 | 0.00 | 0.00 | 0.00 |
| SAMPLE (BOTH) | -0.07 | 0.02 | 0.00 | 0.00 | 0.00 | 0.00 |
| SAMPLE (NAIVE) | -0.05 | 0.01 | 0.00 | 0.00 | 0.00 | 0.00 |
| FAILSAFE (ADAPTIONPENALTY) | -0.06 | 0.02 | 0.02 | 0.04 | 0.00 | 0.00 |
| FAILSAFE (NAIVE) | -0.07 | 0.03 | 0.10 | 0.10 | 0.00 | 0.00 |
| BASELINE (NAIVE) | -0.24 | 0.37 | — | — | 0.20 | 0.40 |
| MASKING (NAIVE) | -0.15 | 0.16 | 0.14 | 0.26 | 0.00 | 0.00 |
| **SAC** | | | | | | |
| PROJECTION (NAIVE) | -0.20 | 0.01 | 0.52 | 0.02 | 0.00 | 0.00 |
| PROJECTION (ADAPTIONPENALTY) | -0.19 | 0.00 | 0.49 | 0.01 | 0.00 | 0.00 |
| PROJECTION (SAFEACTION) | -0.19 | 0.00 | 0.49 | 0.01 | 0.00 | 0.00 |
| PROJECTION (BOTH) | -0.20 | 0.01 | 0.49 | 0.01 | 0.00 | 0.00 |
| SAMPLE (NAIVE) | -0.15 | 0.01 | 0.05 | 0.00 | 0.00 | 0.00 |
| SAMPLE (ADAPTIONPENALTY) | -0.15 | 0.01 | 0.05 | 0.00 | 0.00 | 0.00 |
| SAMPLE (SAFEACTION) | -0.15 | 0.01 | 0.05 | 0.00 | 0.00 | 0.00 |
| SAMPLE (BOTH) | -0.16 | 0.02 | 0.05 | 0.00 | 0.00 | 0.00 |
| FAILSAFE (NAIVE) | -0.21 | 0.01 | 0.17 | 0.02 | 0.00 | 0.00 |
| FAILSAFE (ADAPTIONPENALTY) | -0.17 | 0.01 | 0.04 | 0.00 | 0.00 | 0.00 |
| BASELINE (NAIVE) | -0.88 | 0.02 | — | — | 0.96 | 0.03 |
| MASKING (NAIVE) | -0.14 | 0.02 | 0.00 | 0.00 | 0.00 | 0.00 |

Note: — indicates that there is no intervention rate for the baselines as they don't implement a safety verification.

