# OpenReview forum: "Provably Safe Reinforcement Learning: Conceptual Analysis, Survey, and Benchmarking"
_TMLR — Accepted by TMLR_

### Review · Reviewer_9Fsx · 2023-07-31

**Summary Of Contributions:**

The paper proposes a taxonomy of action selection methods for provably safe RL, comprehensively categorizing existing methods according to this taxonomy. In particular, the types of methods are action replacement, action projection, and action masking. The paper also discusses the different design choices for learning that must be made in light of each action selection method. The experimental results compare certain action selection methods. The paper also discusses practical insights that could prove useful for implementing these methods.

**Audience:**

Yes

**Broader Impact Concerns:**

N/A.

**Claims And Evidence:**

Yes

**Requested Changes:**

The following are the most significant requested changes.
   * Strengthen experimental section in line with my comments above. Either a more thorough experimental evaluation of action selection methods overall or a more focused evaluation of the properties of a few select action selection methods could work.
   * Need a more organized discussion of the benefits of this taxonomy for future work

At the moment, I am leaning towards rejection of the paper unless the above are addressed. Some other issues below are lower priority, but which would greatly improve the paper in my opinion.

   * Improve discussion of related work on safe RL in line with my comments above
   * Include a discussion of the limitations of the present safe RL framework
   * Learning tuples should maybe be its own separate section since it’s relevant for all of the action selection methods
   * Hard to compare rewards of baselines w/ provably safe methods in fig 2; maybe group the lines different or make a separate plot


EDIT: All of my concerns have now been addressed by the authors and I have changed my score

**Strengths And Weaknesses:**

Strengths
   * Overall goal of identifying different action selection methods is good
   * Clear presentation of taxonomy; seems potentially useful as well. E.g., pointing out the relative lack of work on action masking in some applications and for continuous actions
   * Discussion of learning tuples illuminates different design choices and potential consequences
   * Methodology for lit review seems good


Weaknesses
   * Experimental section is quite weak
      * Overall goal is either too strong (“The theory and experiments confirm that provably safe RL methods are always safe, while the baselines still violate the safety results even after the reward converged.”) or unclear
      * Small sample size so can’t give strong recommendations
         * Not a lot of algorithms evaluated
         * Not a lot of environments
         * Hard to make claims about the categories because only a few algorithms are evaluated; claims are only particular to the algorithms
         * Not many seeds used
      * Hard to interpret results of fig 3
         * “A notable result of our evaluation is that using the adapted action…” -> unclear to me from plots
         * Error bars are large
         * Too much going on, potentially too many colours
   * Relatively weak discussion of safety in the related work
      * Safety in the context of this work is with respect to the objective function, but it is often difficult in practice to get that objective function
      * Should talk about reward learning (https://arxiv.org/abs/1706.03741), reward hacking (https://openreview.net/forum?id=yb3HOXO3lX2), goal misgeneralization (https://arxiv.org/abs/2105.14111)
   * Missing a discussion of the limitations of the safe RL framework
      * For example, can the constraint function take into account safe and unsafe trajectories since the constraint is only for a given state and action?
      * Maybe whether a state is desirable or not depends upon the states that have previously been visited; is this accounted for in the MDP formulation?
   * Scattered discussion of the benefits of this taxonomy
      * Not clear that there are enough clear and strong future research directions identified. E.g., should there be more action masking work, and why?

---

> ### Author Response · Authors · 2023-08-20
> **Not a lot of environments**
>
> Regarding the low number of environments tested: it is common practice that the number of environments tested in provably safe RL works is fairly low, as designing the safety function for each environment requires a significant amount of work. In our case, this is exacerbated due to the fact that all three provably safe RL methods have to be comparable, i.e., the resulting safe action sets should have the same size. We believe that we selected two appropriate and widely used systems in the field and are, to the best of our knowledge, the first to provide a comprehensive benchmarking between provably safe RL approaches. To achieve such a comprehensive benchmarking, discretizing the action space has to lead to an action space for which the agent still can find a good policy, and the provably safe action set needs to be convex due to the current limitations of continuous action masking. In addition, the system noise present in our benchmarks exceeds the complexity of many comparable works in table 1. Some presented research would already struggle with the six state dimension of the quadrotor benchmark. To support future research, we will publish our code on CodeOcean (currently submitted as supplementary material), which could be the basis for a provably safe RL benchmarking suite.
>
> To further clarify why we selected the presented benchmarks, we want to exemplify the common limitations for such more complex benchmarks in a human-robot environment. We created a realistic human-robot environment to test the transferability of our benchmark findings to more complex settings. In this environment, a six degree of freedom robotic manipulator has to reach a given goal position, but a human obstructs the straight path to the goal. The action space is three-dimensional (desired Cartesian position of the end effector), and the state space is 11-dimensional. In preliminary tests, we compared action projection and replacement with each other for continuous PPO and the results for five random seeds match the results of figure 2 (*naive* learning tuple). However, we cannot comprehensively compare this benchmark for several reasons.
>
> - We cannot include action masking as the continuous action masking approach proposed in this work is not applicable.
> - We tested discretizing the actions, although not common for robotic manipulation, but using a discrete action space showed very poor performance in preliminary tests.
> - An hypothetical comprehensive comparison would be computationally infeasible due to the high-fidelity simulation. Specifically, one training run takes up to 4 hours (with a distributed implementation, i.e., 10 threads on a machine with an AMD EPYC 7742 2.2 GHz processor and 1024 GB of DDR4 3200 MHz memory), so 650 runs would hypothetically take 108 days.

---

> > ### Comment · Reviewer_9Fsx · 2023-08-22
> >
> > Thanks! This response and the changes to section 4 have addressed my concerns on this front.

---

> ### Author Response · Authors · 2023-08-20
> **Overall goal is either too strong or unclear**
>
> We adjusted the following statements:
>
> - Section 1 page 1:
> ”Our experimental evaluation confirms that all three provably safe RL types guarantee safety and shows that action replacement exhibits the most robust performance across all tested environments and on the investigated RL algorithms.”
> To:
> ”We, furthermore, evaluate the methods experimentally with the three main findings that all provably safe RL methods are indeed safe, that action replacement performs best on average over five tested RL algorithms, and that adding a penalty to the reward when using the safety function further improves performance.”
> - Section 4.3 page 14:
> ”The safety violations for the baselines in figure 2b show that safety can only be guaranteed when using provably safe RL.”
> To:
> ”The safety violation evaluation of the baselines in figure 2d shows that the baseline algorithms fail to guarantee safety during training in the 2D quadrotor stabilization task. All provably safe RL algorithms guarantee safety as expected following from the conceptual analysis in section 2.”
> - Section 4.3 page 14:
> “The comparison between the three provably safe RL categories action replacement (sample), projection, and masking in figure 2c shows that replacement is the most stable and high-performing method.”
> To:
> ”The reward comparison shows that action replacement performs better than action projection and masking on average.”
> - We further changed the discussion on the impact of learning tuples following the updated results of Figure 3. The paragraph now reads (page 15):
> “We evaluate the impact of different learning tuples on the performance and safety activity averaged over all five RL algorithms in figure 3. When action masking is used, only safe actions can be sampled, i.e., only the *naive* tuple is meaningful; so we omit action masking from this evaluation. For both action replacement and projection the *adaption penalty* tuple leads to the highest performance and lowest safety intervention rate, even outperforming the average over the baselines. In action projection, the *naive* tuple performs significantly worse than in action replacement. The *safe action* and *both* tuple seem to be only beneficial when using action projection and decrease performance when using action replacement.”
> - Section 5 page 15:
> From:
> "The theory and experiments confirm that provably safe RL methods are always safe, while the baselines still violate the safety results even after the reward converged.”
>
>     To:
>     ”Our experiments confirm the theoretical statement that provably safe RL methods are always safe when assumption 1 is fulfilled.”

---

> > ### Comment · Reviewer_9Fsx · 2023-08-22
> >
> > Thanks, looks good to me

---

> ### Author Response · Authors · 2023-08-20
> **Experimental improvements**
>
> We integrated your other comments and improved our experimental evaluation and its presentation as clarified now in detail:
>
> - We included soft actor-critic (SAC) as another RL algorithm on both environments and increased the number of random seeds from 5 to 10. With these changes, our evaluation now comprises 1300 RL training runs. We believe these changes improved the strength of the foundation for our recommendations.
> - Previously, we showed the tuple comparison results only on discrete PPO, leaving the reader with the tedious task of comparing the results for other algorithms in the Appendix. This was a bad design decision, which the reviewer correctly pointed out (”Not a lot of algorithms evaluated”). We now summarize the main results over all five RL algorithms (TD3, SAC, PPO continuous, PPO discrete, and DQN) combined to make more general statements. The new Figure 3 compares the performance of the different learning tuples averaged over all five algorithms. We also removed the comparison between the sample and failsafe strategy for action replacement in this figure as it could confuse the reader.
> - After closer inspection of the training curve distribution on 10 random seeds, we concluded that the training runs are not normally distributed, so the standard deviation is not a suitable metric to report. Therefore, we decided to report how certain we are in the mean metric with the 95% interval obtained via bootstrapping. This method was recently recommended in “Empirical Design in Reinforcement Learning” by Patterson et al. [https://arxiv.org/abs/2304.01315 , Chapter 2.5]. This interval only depicts our confidence in the mean metric and is, therefore, often smaller than the previously reported standard deviation metric. In addition to the improved statistical interpretation, we find the resulting plots better readable.
> - We added a green dashed line for the average reward performance of all baseline algorithms to all reward plots. This should improve comparability between the results.

---

> > ### Comment · Reviewer_9Fsx · 2023-08-22
> >
> > Thanks, these are really good improvements!

---

> ### Author Response · Authors · 2023-08-20
> **Relatively weak discussion of safety in the related work**
>
> We hope that we understood you correctly that a discussion is missing why it is hard or close to impossible to guarantee safety through the reward in RL. There is the “reward is enough” hypothesis in RL, which is currently under a lot of discussion, and we believe that briefly discussing the safety aspects of this hypothesis in this paper adds a lot of benefit for RL researchers with less background in formal methods. Therefore, we added the following paragraph to Section 1.1:
>
> “Soft constraint approaches consider safety directly in their optimization objective. Here, the agent can explore all actions and states regardless of safety. Thus, these methods can be unsafe during training, especially in the beginning, but converge to a safer policy without formal safety guarantees after sufficient training steps. The simplest way to inform an RL agent about safety constraints is through its reward function. Despite its elegance, the reward function approach has many potential pitfalls. First, the reward function might be ill-defined, either from manual tuning or when learned from human input. When manually defined, the reward function might overlook certain features or fine details, leading to a hackable reward (Skalse et al., 2022) from which the agent learns an unsafe behavior. Learning the reward function from human feedback (Christiano et al., 2017) is also error-prone because communicating safety constraints alongside performance metrics is hard for sparse, non-linear, conditional, or seldom occurring constraints. Second, even if the reward function is defined correctly, the trained policy is not guaranteed to be safe, as it was shown by Packer et al. (2019) that RL agents struggle with out-of-distribution states during deployment. Third, the agent might learn to perform actions safely but ignore the task objective due to goal misgeneralization (Langosco et al., 2022).”

---

> ### Author Response · Authors · 2023-08-20
> **Missing a discussion of the limitations of the safe RL framework**
>
> We agree with the reviewer that a discussion about the limitations of our provably safe RL framework improves the quality of our work. Hence, we now include the following paragraph in our discussion:
>
> ”**Limitations of provably safe RL** There are limitations for provably safe RL that follow from the conceptual analysis in section 2. Most importantly, safety has to be decidable, i.e., there must be a safety function $\varphi(s,a)$, which fulfills assumption 1. For this safety function, there is system knowledge necessary, and especially for systems with a high number of continuous state variables, the safety function is potentially hard to retrieve. Additionally, safety guarantees are strongly tied to the safety function. If the safety function provides complex guarantees (e.g., temporal logic), it is usually computationally more expensive than for simpler guarantees (e.g., system stays within safe state set). Second, safety can only be decided if the state of the system is correctly observed within noise bounds. Thus, for a fully provably safe system, the perception module also needs to be verified such that it provides observations that are correct within the noise bounds. Third, there is often a trade-off between safety and performance since for many tasks these two objectives are only partially aligned. For example, if a car drives faster it reaches its destination earlier but collisions are more difficult to avoid with higher speed. Since provably safe RL only allows safe behaviour, there is no such trade-off as safety is always prioritized over performance. Thus, the safety function $\varphi(s,a)$ cannot be too conservative since then the agent would only perform actions that are safe for an infinite time horizon, e.g., often the stand still action. Lastly, comparing provably safe RL approaches is challenging as we need to define a safety function $\varphi(s,a)$ that is efficiently usable by different approaches. Furthermore, the notion of safety is usually application-specific, so different application-specific approaches are hard to compare. We provide the first comparison of provably safe RL on two common continuous stabilization tasks, but further research is necessary to make stronger claims about the best provably safe RL approach.”
>
> Regarding your concern of unsafe trajectories: the constraint function $\varphi(s,a)$ (introduced in (1)) evaluates trajectories for their safety. Although, only the current action is verified, the safety function can take into account a longer time horizon (e.g., all possible trajectories after the first action). This is the case for most safety verification methods, including MPC, CBF, HJI, set-based reachability analysis, temporal logic, and dL. We believe that the sentence after assumption 1 (page 4) discusses this point concisely:
>
> "With assumption 1, it is ensured that only provably safe states $s^\varphi \in \mathbb{S}_\varphi$ can be reached when starting from any $s^{\varphi}_0$ and taking only provably safe actions thereafter.”
>
> We discuss several concrete implementations of $\varphi$ in the related work section.

---

> ### Author Response · Authors · 2023-08-20
> **Discussion of the benefits of this taxonomy for future work**
>
> We removed the future work from the Conclusion and reformulated it to two new paragraphs at the end of the Discussion (page 17):
>
> ”**Future research based on proposed taxonomy** Most action projection approaches discussed in section 3 project the RL action on the edge of $\mathbb{A}\_\varphi$. In our experiments, we encountered two negative side effects related to this action projection implementation: First, the projection to the edge of $\mathbb{A}\_\varphi$ often leads to a relatively small $\mathbb{A}\_\varphi$  in the next RL step, quickly resulting in a very small set $\mathbb{A}\_\varphi$  after a few unsafe RL actions. Second, small floating point errors can cause unsafe actions and must be handled. Therefore, future action projection research should investigate the objective function of (3) to obtain more stable training. Action masking is a promising technique but has mainly been used in grid world environments and games, e.g., the Atari benchmark Huang & Ontañón (2022), with discrete action spaces so far. Our proposed continuous action masking approach only applies to specific environments and performed well for the pendulum but showed mixed results for the 2D quadrotor. Thus, future research should investigate ways to extend continuous action masking to general convex or even non-convex $\mathbb{A}\_\varphi$  representations in order to improve its applicability to more complex benchmarks and reduce the conservativeness of $\mathbb{A}\_\varphi$ . Additionally, it should be investigated if the agent should be informed about the reduction of the action space through masking. This could result in improved convergence and an agent that is less dependent on the safety verification, similar to the effect of the adaption penalty tuple for action replacement and action projection. The evaluation of the considered benchmark shows that action replacement performs better than action projection and masking as discussed previously. However, it is still unclear if the replacement strategy $\psi(s)$ is important for the convergence and performance of the agent, especially when applied to more complex tasks. Thus, future action replacement research should empirically and theoretically investigate this question.
>
> **Improving applicability of provably safe RL** Despite the promising previous work discussed in section 3, there are few works on high-dimensional non-linear systems and limited real-world applications. We suggest five major factors where future research would improve applicability. First, some approaches need to be computationally more efficient to be real-world applicable. Especially the computational efficiency of verification methods is relevant and should be improved as discussed previously. Second, we observe that the type of learning tuple used has a significant influence on the performance of the agent for some RL algorithms. Also, there is little theoretic research on how provably safe RL approaches influence convergence to an optimal policy. More empirical and theoretical research on the convergence effects of provably safe RL and its learning tuples would support effective and efficient implementation of the algorithms. Third, to evaluate new provably safe RL approaches common benchmarks are necessary. Additionally, the three action correction strategies should be compared on more complex benchmarks to clarify if our observations can be extended to them. Such benchmarking would make the provably safe RL research more comparable, would ease starting research on provably safe RL, and would provide more evidence to decide for the best-suited provably safe RL approach. Fourth, recent work shows a low variety of safety specifications, mainly comprising stabilization and reach-avoid specifications. Contrary, real-world safety is more complex, e.g., traffic rules such as waiting at a red light and safely but quickly moving at a green light. Finally, provably safe RL requires expert knowledge on verification methods. Future research could mitigate this through modular and automatic approaches, where fewer engineering decisions are necessary, and more parameters can be tuned automatically. With these advances, provably safe RL could bring the best of RL and formal specifications together towards RL methods that require as little expert knowledge as necessary and provide formal guarantees for complex safety specifications to achieve reliable and trustworthy cyber-physical systems.”
>
> ## Comment 5:
> Learning tuples should maybe be its own separate section since it’s relevant for all of the action selection methods
>
> ## Answer to comment 5:
> We moved the discussion about learning tuples to its own subsection (section 2.4, page 8).

---

> ### Comment · Reviewer_9Fsx · 2023-08-22
>
> The authors have addressed my concerns to a sufficient degree that I now feel comfortable accepting the paper.

---

> > ### Author Response · Authors · 2023-08-31
> >
> > Dear Reviewer,
> >
> > Thanks a lot for your positive response. Your review was very helpful for us to improve our paper. We really appreciate your effort.

---

### Review · Reviewer_CFAn · 2023-07-31

**Summary Of Contributions:**

### High Level Overview

Reinforcement learning has been used in different areas and applications such as robotics, autonomous systems, and games.  However, in safety-critical applications it is important to design and develop RL algorithms with safety guarantees.  For example, consider using an RL algorithm for autonomous driving. Whether the algorithm is trained using simulations or with historical data, the trained policy must abide by road constraints and ensure safety of the vehicle and its surroundings.  Vanilla RL algorithms are typically set-up to explore potentially unsafe regions focused solely on reward maximization.  In contrast, safe RL algorithms limit the exploration and execution to only consider “safe” parts of the state-action space.  At a high level, the authors here provide a survey of “provably safe” RL algorithms, where provably safe refers to the fact that safety constraints are satisfied both during training and also during test time.  They introduce a categorization of existing provably safe RL methods into three types: action replacement, action projection, and action masking.  Lastly, they provide experimental evaluation to confirm that existing algorithms from the safe RL field indeed satisfy the safety constraints, and provide insights into which algorithm framework should be used in a given application area.

### Main Model

RL is build upon the framework of an MDP, defined as a tuple $(S,A,T,r,\gamma)$ where $S$ is set of states, $A$ set of actions, $T$ the transition distribution, $r$ the rewards, and $\gamma$ the discount factor.  For provably safe RL it is required that safety of state-action pairs are verifiable, hence the authors assume there is a safety function $\phi(s,a)$ which outputs $1$ if the state-action pair is verified to be safe, and zero otherwise.  It is clear that executing $\phi(s,a)$ or even the knowledge of such a function requires some prior knowledge.  However, in settings without some knowledge of the dynamics that provably safe RL is not feasible.  Lastly, the authors make the following key assumption that there is at least one provably safe initial state, and that for all safe states there exists at least one safe action.

### Summary

Action Replacement: The first approach the authors discuss to ensure safety is to replace any unsafe action outputted by the agent with a safe one before it gets executed.  Indeed, if the policy $\pi$ proposes an action $a$, then if $\phi(s,a) = 1$ then $a$ is taken.  Otherwise, the authors assume there is a replacement policy $\psi(s)$ which always guarantees to output a provably safe action, and so $\psi(s)$ is instead taken.  Note that $\psi(s)$ can either be sampled randomly from the safe set, or using a failsafe controller.

Action Projection: The second approach in the literature is action projection. Here, if the action proposed by the policy $\pi$ is unsafe, then the actual action taken solves the following optimization problem, whereby you minimize $d(a, \tilde{a})$ subject to $\phi(s,\tilde{a}) = 1$, i.e. project the action $a$ to the set of feasible actions and find the closest point.  This can be formulated using either control barrier functions or using additional domain knowledge.

Action Masking: The last approach limits the action space $A$ that is fed into the RL algorithm.  Indeed, assume you are given a function $\eta(s)$ that outputs a set of provably safe actions.  Then the policy $\pi$ which is learned is informed by both $\eta(s)$ (dictating the actions available in that state) as well as the current state $s$.  While this is straightforward in discrete action spaces, the authors additionally outline its implementation with continuous action spaces.

### Experimental Results

Lastly, the authors complement their literature review with experimental results.  They compare “unsafe” baseline algorithms to their “safe” counterparts on two tasks (a pendulum and quadrotor task).  The results highlight that the safe algorithms indeed satisfy the safety constraints both during training and during test.  However, they additionally notice that action replacement is the most stable and high-performing method.


**Audience:**

Yes

**Broader Impact Concerns:**

The work provides a general purpose summary of proposed RL algorithms for safety-critical applications.  The authors provide no new algorithms or implementations, mostly summarize existing algorithms and ideas in the field.  Lastly, the authors mention that researchers utilizing safe RL algorithms should adhere to ethical standards within their particular context.

**Claims And Evidence:**

Yes

**Requested Changes:**

### Requested Changes
- The authors should provide a discussion on the distinction between implementing these algorithms in an offline versus an online setting. This is briefly discussed on page 5 but not elsewhere.  There are important considerations between the feedback, adjusting for importance sampled weights for the behaviour policy, etc, which should be included.
- The experimental results seem to provide no statistical insights between the performance of the different algorithms (especially with the inclusion of the confidence intervals).  This makes it difficult to appreciate one of the key findings that action replacement is the most stable and high-performing method (when in reality the performance is statistically indistinguishable).  Potentially this could be ameliorated by running more experiments on different areas.  However, as written in Section 4.3 it is difficult to derive any actionable insights form the experiments other than safe RL methods are indeed safe.  The first paragraph in page 14 is helpful to this, and could be used as a guide for the rest of the section.
- Action masking for continuous state-aciton pairs seems under-developed. The authors only consider a straightforward implementation of it, without any extensions to arbitrary convex action spaces, or tools for potentially computing a "safe" bounding box within the safe set $A_\phi$ using tools from optimization for generating non-axis parallel transformations.

### Questions
- What is the reason for the performance gap on the figure in page 13 between “safe” and baseline algorithms? Does this imply that the optimal policy does not necessarily follow a safe trajectory? Issues in convergence for the provably safe RL algorithms?
- Do you have any thoughts on extensions for action masking to other continuous action space?

### Minor Comments
- Language in the first paragraph is a bit over-the-top and could be toned down
- Third paragraph on page one is a nice succinct summary of the main contributions in the paper
- Under Section 1.1 should mention that consider “provable” as only “hard” safety constraints and highlight that it means safe during both learning and execution
- Loved the diagram in Figure 1 - helped highlight the different approaches and taxonomy of safe RL algorithms proposed
- Page 3 under “safety of system” can highlight how can incorporate aggregate or cost constraints by augmenting the state space (e.g. considering budget)
- Safe action set A_s(s) seems poorly defined and used a couple places throughout the paper. Recognize that you are trying to highlight that the safety function $\phi$ doesn’t need to necessarily be perfect and allow for some type 2 errors (i.e. the state is safe but it is predicted by $\phi$ to be unsafe) but no type 1 errors.  Think that this could be clarified and explained a bit more, since the notation is used later in in the discussion with the taxonomy
- First paragraph under section 2.1 discussing the evaluation of $\phi$ or over-approximating the set of safe states could be moved to the previous section
- Top of page 7 in equation (9) should be defined by $\eta(s)$
- Enjoyed discussion on page 7 starting with “Since the action spaces for RL …”
- Results in Table 2 seemed very interesting, highlighting real-world high fidelity simulations. Would have enjoyed some more discussion into the practical insights from these papers in section 3
- First paragraph on page 14 was great - provided nice actionable advice


**Strengths And Weaknesses:**

### Strengths
- The authors provide a thorough literature review and classification of the existing work on safe RL algorithms.  This helps summarize the entire field, develop insights for implementing the algorithms in new domains, and potential ideas for modifying existing methods to fit the given application.
- Based on the experimental results, on page 14 the authors provide a succinct summary on their experience implementing provably safe RL algorithms to the domains considered in the paper. This helps provide actionable insight for implementing the algorithms in practice.

### Weaknesses
- The experimental results seem limited and hard to derive statistical significance on the comparison between the different methods.
- The discussion on implementing action masking in continuous state-action spaces is under-developed and could be expanded to more general convex action spaces.

---

> ### Author Response · Authors · 2023-08-20
> **Reply to requested changes of Reviewer CFAn**
>
> Thank you for your valuable comments and suggestions. We especially appreciated that you highlighted some specific parts that you enjoyed. We incorporated your comments. Please find the replies to your change requests below.
>
> ### Requested Changes
>
> **Comment 1:** The authors should provide a discussion on the distinction between implementing these algorithms in an offline versus an online setting. This is briefly discussed on page 5 but not elsewhere. There are important considerations between the feedback, adjusting for importance sampled weights for the behavior policy, etc, which should be included.
>
> **Answer:** We added the paragraph **Online vs. offline implementation** in the discussion (end of page 16) to clarify this.
>
> **Comment 2:** The experimental results provide no statistical insights between the performance of the different algorithms (especially with the inclusion of the confidence intervals). This makes it difficult to appreciate one of the key findings that action replacement is the most stable and high-performing method (when in reality the performance is statistically indistinguishable). Potentially this could be ameliorated by running more experiments on different areas. However, as written in Section 4.3 it is difficult to derive any actionable insights form the experiments other than safe RL methods are indeed safe. The first paragraph in page 14 is helpful to this, and could be used as a guide for the rest of the section.
>
> **Answer:** We first ran the experiments on 5 additional random seeds, so that our total number of runs per configuration is now 10. After closer inspection of the distribution of runs, we noticed that the data is not normally distributed, as often the case in RL. This makes the standard deviation a misleading statistic to report. Therefore, we now report how confident we are in the reported mean. We do that by calculating the 95% confidence interval in the mean metric using bootstrapping, as proposed in “Empirical Design in Reinforcement Learning” by Patterson et al. [https://arxiv.org/abs/2304.01315 , Chapter 2.5]. Our statements are much better supported statistically with regard to the mean metric. Additionally, we clarified the selection of our benchmarks in the experimental section and restructured the discussion to more clearly present our experience and intuition with provably safe RL benchmarking.
>
> **Comment 3:**  Action masking for continuous state-aciton pairs seems under-developed. The authors only consider a straightforward implementation of it, without any extensions to arbitrary convex action spaces, or tools for potentially computing a "safe" bounding box within the safe set using tools from optimization for generating non-axis parallel transformations.
>
> **Answer:**
>
> Thank you for your comment and question. We agree that our continuous action masking implementation is relatively straightforward and would like to argue that the main reason for introducing this theory was to provide the comprehensive benchmarking. We believe that a more developed formulation of action masking would merit a new article since it would be necessary to present the verification method that provides the provably safe action set in-depth, discuss the theoretical guarantees, and investigate the effect on the learning process. Thus, this is out of scope for this survey and benchmarking paper, which we now clarify in section 2.3. and in the discussion.
>
> We further agree with the reviewer that adding an initial concept for an advanced continuous action masking to our work improves its quality. Hence, we added the following explanation to section 2.3 (mid of page 7):
> “Note that this implementation approximates the provably safe action set with an interval set which can be conservative. To overcome this limitation of this first formulation of continuous action masking, more complex set representations, such as the zonotopes (Althoff et al., 2021), for the action spaces ($\mathbb{A}$ and ${\mathbb{A}}{{\varphi}}$) in combination with solving an optimization problem that maximizes the size of ${\mathbb{A}}{{\varphi}}$ could be investigated. A less sophisticated yet possibly effective approach is searching for a good latent interval representation of and transformation to ${\mathbb{A}}{{\varphi}}$ by applying principal component analysis to a set of ${\mathbb{A}}{{\varphi}}$ for different states as a pre-computing step.”

---

> ### Author Response · Authors · 2023-08-20
> **Reply to questions of Reviewer CFAn**
>
> ### Questions
>
> **Question 1**: What is the reason for the performance gap on the figure in page 13 between “safe” and baseline algorithms? Does this imply that the optimal policy does not necessarily follow a safe trajectory? Issues in convergence for the provably safe RL algorithms?
>
> **Answer:** Yes, safety and performance do not necessarily align. For example, for the 2D quadrotor, the reward penalizes deviation from the optimal state $s^{\ast}$ and actions different from $a_{\text{min}}$, but safety is defined by the safe set $\mathbb{S}\_{\varphi}$, for which there exists an action that keeps the agent in $\mathbb{S}\_{\varphi}$. So there might be actions that bring the agent faster to the optimal state $s^{\ast}$ whose execution can be safe or unsafe depending on the noise. Thus, the baseline agents learn to possibly violate safety (see Figure 2 (a)) but approach the optimal state faster in contrast to the safe agents that cannot execute such potentially unsafe actions. Since these actions would not be verified safe and are thus not available for the provably safe agents. In summary, the optimal policy (with respect to the reward) does not necessarily follow a safe trajectory.
>
> There is literature discussing how provably safe RL approaches affect convergence, e.g., Hunt et al. (2021), Huang & Ontañón (2022), Gros et al. (2020). However, most works disregard this discussion as, in practice, the action corrections often have little effect on convergence and can be viewed as a disturbance. This is also one motivation why we compared the different learning tuples. We agree that this topic requires more research and added this future work suggestion to the discussion.
>
> **Question 2**: Do you have any thoughts on extensions for action masking to other continuous action space?
>
> **Answer:** Please refer to our answer to your Requested Changes Comment 3.

---

> ### Author Response · Authors · 2023-08-20
> **Reply to minor comments of Reviewer CFAn**
>
> ### Minor Comments
>
> **Comment 4:** Language in the first paragraph is a bit over-the-top and could be toned down
>
> **Answer:** We revised the first paragraph to make clearer what we mean by provably safe RL and toned it down.
>
> **Comment 5:**  Under Section 1.1 should mention that consider “provable” as only “hard” safety constraints and highlight that it means safe during both learning and execution
>
> **Answer:** We highlighted our definition of provably safe RL in section 1.1. and also tried to clarify it more throughout the text (mainly abstract, introduction, conclusion).
>
> **Comment 6:**  Page 3 under “safety of system” can highlight how can incorporate aggregate or cost constraints by augmenting the state space (e.g. considering budget)
>
> **Answer:** We added the following clarification in the text (page 3): “Note that the state space is often augmented from the classical control state space to a state space that also includes other safety-relevant dimensions, e.g., action space with constraints.”
>
> **Comment 7:**  Safe action set $\mathbb{A}\_s(s)$ seems poorly defined and used a couple places throughout the paper. Recognize that you are trying to highlight that the safety function doesn’t need to necessarily be perfect and allow for some type 2 errors (i.e. the state is safe but it is predicted by to be unsafe) but no type 1 errors. Think that this could be clarified and explained a bit more, since the notation is used later in in the discussion with the taxonomy
>
> **Answer:** We added the clarifying sentences at page 4:
>
> “The safe action set $\mathbb{A}\_s(s)$ includes all safe actions while the provably safe action set $\mathbb{A}\_\varphi(s)$ only includes actions that are verified as safe by the safety function $\varphi(s, a)$. In other words, the safety function possibly returns that an action is unsafe which is indeed safe, while it never predicts truly unsafe actions to be safe.”
>
> **Comment 8:** First paragraph under section 2.1 discussing the evaluation of or over-approximating the set of safe states could be moved to the previous section
>
> **Answer:** We moved the following information to section 2.1 (page 4). ”Conceptually, this is mostly done by over-approximating the set of states that are reachable by taking action $a$ in state $s$, and then validating if the reachable set of states is a subset of $\mathbb{S}\_{\varphi}$.
> We discuss the concrete verification methods used by previous works in section 3.”
>
> **Comment 9:** Top of page 7 in equation (9) should be defined by $\eta$
>
> **Answer:** We clarified the that $\eta:  \mathbb{S} \rightarrow \mathbb{A}$ while $\varphi(s, a): \mathbb{S} \times \mathbb{A} \rightarrow \{0,1\}$. For equation (9), we need the binary encoding of safe and unsafe actions to arrive at the masked policy $\pi\_m$. Thus, discrete action masking can be viewed as an informed drop-out layer so that unsafe actions get probability 0 assigned. We see that using $\eta$ in equation (9) seems inconsistent, so we added an explanation for discrete action masking stating how $\eta$ and $\varphi(s, a)$ relate to each other (top of page 7):
>
> “For discrete actions, the safety of each action is verified in each state using $\varphi(s, a)$ and all verified safe actions are added to $\mathbb{A}\_\varphi(s)$, i.e., $\eta$ iterates over all actions for the current state $s$ with $\varphi(s, a)$ to identify $\mathbb{A}\_\varphi(s)$.”
>
> **Comment 10:** Results in Table 2 seemed very interesting, highlighting real-world high fidelity simulations. Would have enjoyed some more discussion into the practical insights from these papers in section 3.
>
> **Answer:** Thanks a lot for this comment. We felt like extending the discussion on the application-specific papers would extend the section 3 with too much detail. However, we could provide a more in-depth discussion of the application-specific papers in the appendix with a table similar to table 1. This would be a considerable effort, so we would appreciate your opinion on this suggestion.

---

> > ### Comment · Reviewer_CFAn · 2023-08-21
> > **Comments**
> >
> > Thank you for addressing all of my minor comments! I appreciate the effort - and think that in the current version the description of $A_{\phi}(s)$ makes the writing much easier to follow along.
> >
> > In reference to comment 10: a potential in-between (assuming these papers also do a similar ablation study over different safe RL methods) is to add a comment in section 5 if the practical safe-RL algorithms noticed similar finding (that action masking is more efficient than the other provable safe RL algorithms).

---

> > > ### Author Response · Authors · 2023-08-31
> > >
> > > Dear Reviewer,
> > >
> > > Thanks again for your valuable review. It is great to hear that you like our changes and, in particular, the changed description of $\mathbb{A}_\varphi(s)$.
> > >
> > > Regarding comment 10: Thanks for your suggestion. Most application-specific works validate their method against an unsafe RL baseline or on different environment configurations. However, a few include a comparison between different provably safe RL methods.
> > >
> > > - Ceusers et al. (2023) compare fail-safe action replacement and sampling-based action replacement. They observe that both methods have higher initial performance than the unsafe RL baseline, and fail-safe action replacement leads to better performance than the sampling-based version.
> > > - Brosowsky et al. (2021) observe that their masking approach converges slightly faster than action projection.
> > > - Wang (2022) compares the deployment of a discrete action masking approach with her continuous action projection approach. The goal-reaching performance is lower for the discrete action masking approach. However, this could be due to the coarse discretization of the action space in three actions.
> > >
> > > We integrated this into the literature review section of the paper.

---

### Review · Reviewer_xEjt · 2023-08-09

**Summary Of Contributions:**

This paper presents a survey of provably safe reinforcement learning algorithms, and an experimental comparison of some prior approaches on simple simulated control environments. The paper categorizes prior works under three categories: action replacement, action projection, and action masking depending on how safe behavior is guaranteed by the algorithm. The experiments evaluate these different types of algorithms and attempts to provide consolidated insights into which type of algorithm might be suited for a particular type of problem.

**Audience:**

Yes

**Broader Impact Concerns:**

None.

**Claims And Evidence:**

No

**Requested Changes:**

Please refer to the detailed list of weaknesses above. In summary:

1) is it possible to provide any comparisons of how the theoretical guarantees in the prior works differ from each other, and how they can be viewed under a unified lens?

2) is it possible to provide the experimental comparisons in more realistic (simulated / real world) environments, for example the ones described in Brunke et al. 2022?

**Strengths And Weaknesses:**

Strengths

- This survey paper is clear and easy to read, and has detailed coverage of several different prior works.
- The paper attempts to provide unifying perspective on prior works by categorizing them into different groups based on how safety is ensured, and through experiments in consistent environments that provide perspectives on relative strengths/weaknesses of these different classes of safe RL algorithms
- The overall summaries of prior works in terms of algorithm designs and applications in the two big tables are helpful in distilling the differences at a glance.

Weaknesses

- the paper does not provide a lot of insights beyond what prior works already cover (in particular other surveys like Brunke et al. 2022) and the there isn't a lot of interesting conceptual  comparisons beyond the three broad categories.

- the paper does not provide any comparisons of how the theoretical guarantees in the prior works differ from each other, and how they can be viewed under a unified lens. Which papers have better guarantees under certain constraints of the environment needs to come out more clearly. Right now, the comparisons are mostly at the level of algorithm design.

- the motivation of the paper is about practical considerations of why provably safe RL is necessary for real world deployments, but the experimental comparisons are very limiting. I am not sure why a 1D pendulum environment, and a 2D quadrotor environment provide insights for safe RL algorithms that would have the possibility of translating to real world systems (that are typically high dimensional both in terms of observation space and action space) For examples of such environments (in simulation), refer to Fig 5 of prior survey paper Brunke et al. 2022

Broadly, given the above limitations, and with existing prior survey+comparison papers like Brunke et al. 2022, this paper does not provide any interesting insights that are likely to be helpful in developing safe RL algorithms of the future.

---

> ### Author Response · Authors · 2023-08-20
> **Reply to Reviewer xEjt (1/2) - Relevant contributions with respect to Brunke et al.**
>
> Thank you for your review. We appreciate that you found our paper easy to read and liked the literature review tables. We recognize your concern that our work might not be helpful in developing new safe RL algorithms in the light of previous safe RL surveys, such as Brunke et al. 2022 and we would first like to clarify the necessity for our survey. Brunke et al. 2022 only provide a categorization and literature review for a more loosely defined field of control and safe learning for robotics. In contrast, we regard the field of *provably safe RL,* where hard safety guarantees are always fulfilled during both learning and operation. By providing assumptions necessary for *provably safe RL*, we delineate the field from the broader term safe RL. We propose a comprehensive and clear categorization for the field of *provably safe RL* through conceptually analyzing and conducting a systematic literature review in section 3. Most of the papers we surveyed (table 1 and 2) are not cited by Brunke et al. 2022 due to the different scope of their survey, e.g. Könighofer et al. (2020), Hunt et al. (2021), Fulton & Platzer (2018), Gros et al. (2020). Additionally, we are the first to compare these different *provably safe RL* approaches on two benchmarks and five different RL algorithms. Since in previous works, there are different learning tuples used, and there is no comprehensive comparison, we also compare the different options for learning tuples empirically. We believe that this collection of concise classification of provably safe RL methods, systematic literature review and benchmarking clarifies the future research directions as well as provides practical insights for researchers selecting a *provably safe RL* approach. We further would like to discuss how we address your concerns in the revised version of our work.
>
>
>
> **Comment:** the paper does not provide a lot of insights beyond what prior works already cover (in particular other surveys like Brunke et al. 2022) and the there isn't a lot of interesting conceptual comparisons beyond the three broad categories.
>
> **Answer:**  While Brunke et al. 2022 provide a helpful survey for the recent developments of learning and control in robotics, we consider specifically *provably safe RL* where hard safety guarantees are provided. We argue that this field of research is of high importance as many robotic system should only be deployed in the real world if formal guarantees are given. The main advantages of our work over Brunke et al. are:
>
> - In comparison to Brunke et al., we provide the reader with a comprehensive taxonomy how provable safety can be achieved in RL. Brunke et al. only presents the concrete verification methods robust control invariant sets, control barrier functions, Hamilton-Jacobi reachability analysis, and safety certification using model predictive control. They do not discuss the approaches in a way that similarities and differences are distinguishable. Furthermore, our taxonomy is formulated so that other approaches like set-based reachability analysis or action masking are also covered. We believe that future verification methods will also fit in our proposed taxonomy.
> - Our taxonomy covers all previous provably safe RL approaches, so that we also survey significantly more previous work than Brunke et al. Most of the papers we surveyed (table 1 and 2) are not cited by Brunke et al. 2022 due to the different scope of their survey, including important works such as Könighofer et al. (2020), Hunt et al. (2021), Fulton & Platzer (2018), and Gros et al. (2020).
> - Brunke et al. do not compare the different provably safe RL approaches experimentally. We believe that our experimental evaluation and subsequent discussion gives the reader a good intuition for the selection of a fitting provably safe RL method.
>
> We clarified the differences to Brunke et al. 2022 in section 1.1. and improved the introduction to clarify the term provably safe RL.

---

> > ### Comment · Reviewer_xEjt · 2023-09-06
> >
> > Dear authors - Thanks for the helpful clarifications regarding similarities and differences with respect to Brunke et al. 2022. The details in the revised paper regarding this are also much clearer now, and will be helpful for readers.

---

> ### Author Response · Authors · 2023-08-20
> **Reply to Reviewer xEjt (2/2) - Verification methods and complex environments**
>
> **Comment summary:**
> Is it possible to provide any comparisons of how the theoretical guarantees in the prior works differ from each other, and how they can be viewed under a unified lens?
>
> **Answer:** The hard safety guarantees are always fulfilled by all provably safe approaches (also validated by our experiments). So there is not really a better guarantee if two approaches provide the same. However, the verification method usually determines what type of systems can be verified, what specifications can be considered, and if the method is computationally feasible for high-dimensional systems. We agree with the reviewer that these points were not sufficiently discussed in the initial submission. Hence, we now discuss the applicability of the verification methods in the related work.
> - LTL model checking: The advantage of online model checking is that linear temporal logic constraints can be guaranteed for general nonlinear systems and the online complexity is linear in state dimensions. However, the method is only applicable to discrete state spaces, constructing the automaton offline has exponential complexity in the state dimension, and the online checking complexity also increases exponentially with the formula length (Baier & Katoen, 2008)
> - HJI reachability analysis: Hamilton-Jacobi-Isaacs reachability analysis can verify reach-avoid problems with arbitrary non-convex sets (Althoff, 2015) and disturbances that stem from a compact set. However, constructing the safe set offline scales exponentially in complexity with the number of states, which makes Hamilton-Jacobi-Isaacs reachability analysis infeasible for systems with more than four states (Chen & Tomlin, 2018).
> - Set-based reachability analysis: Set-based reachability analysis is applicable to reach-avoid problems for general nonlinear systems with uncertainties in the initial state, system and input disturbances, and time delays as long as they stem from a compact set (Althoff et al., 2021). The reachability analysis has polynomial offline and online complexity in the system dimension for most set representations as discussed by Althoff et al. (2021). However, in comparison to Hamilton-Jacobi-Isaacs reachability analysis, set-based reachability analysis cannot handle arbitrary non-convex sets but depends on specific set representations.
> - Control-barrier function: When CBF are given, this approach is advantageous as solving the optimization problem in (3) is polynomial in system dimension. Theoretically, the CBF approach is applicable to general control-affine systems with disturbances from a compact set for reach-avoid specifications. However, guessing the CBF is not trivial and synthesizing them can be exponentially complex in system dimension (Ames et al., 2019).
> - Model-predictive control: Despite its applicability to nonlinear systems, MPC are mostly used for linear systems when the controlled plant is fast in relation to the verification (Zeilinger et al., 2014). MPC is applicable to reach-avoid problems with measurement and state disturbances.
>
> **Comment summary:**
> Is it possible to provide the experimental comparisons in more realistic (simulated / real world) environments, for example the ones described in Brunke et al. 2022?
>
> **Answer:**
>
> We added an explanation of our choice of benchmarks in section 4. The main reasons for selecting these two benchmarks are:
>
> - Comparability: these two benchmarks are commonly used in the literature and provide good comparability of all provably safe RL approaches.
> - Real-world implications: Despite their low-dimensionality, our results are likely transferable to real-world systems, since real-world complexity is often reduced in practice by using lower-dimensional abstract models and an additional noise term. Conformance checking techniques (https://mediatum.ub.tum.de/doc/1554435/965803.pdf and https://doi.org/10.1109/TRO.2023.3277268    ) can then guarantee that the abstract model incorporates the real-world behavior of the system.
> - Realisticity: Our environments include system noise, which is not the case for many of the experiments in the related work listed in table 1. Hence, we believe that our experiments are sufficiently realistic compared to common benchmarks in the field.
> - The environments shown in Fig. 5 of Brunke et al. 2022 are exemplary experiments which are not trivial to reproduce, or even develop new provably safe RL methods for. We believe that our experiments can serve as a reproducible benchmark since we provide our code in CodeOcean.
>
> We discuss the possibility of incorporating an additional environment in this comment: https://openreview.net/forum?id=mcN0ezbnzO&noteId=UDZ3ozhLN7

---

> > ### Comment · Reviewer_xEjt · 2023-09-06
> >
> > Dear authors - thanks for the response regarding theoretical guarantees and experimental environments. The clarifications are helpful, and yes I mostly meant comparison of whether the approaches in respective papers are practically feasible for reasonably high-dimensional systems that are likely to be relevant for real-world safety critical applications. For the experimental comparisons, I am unfortunately not convinced by the arguments. Conformance checking techniques etc. referenced in the response do not seem to be generally applicable for robotic systems (even ubiquitous real world systems like robot arms, quadrupeds, bipeds etc.) I agree that the simpler 1D and 2D environments considered in the paper can serve as reproducible benchmarks, but not very useful ones, unfortunately - researchers trying to develop provably safe RL algorithms for real-world applications are most likely not looking to adopt these environments. It would be helpful if future iterations of the paper could take this into account. However, since my other comments have been reasonably addressed, I would be more leaning towards accept for the paper.
> >
> > Thanks again for the rebuttal responses!

---

### Author Response · Authors · 2023-08-20
**Revision summary**

We thank all reviewers for their valuable comments and suggestions and the editor for organizing the review. We are happy that our proposed taxonomy is viewed as valuable and meaningful (in particular by reviewers “9Fsx” and “CFAn”). We especially appreciated that all reviewers liked the systematic literature review and, in particular, the summarizing tables. We believe that within safe RL, provably safe RL is a key subfield as it provides hard guarantees that are crucial for all safety-critical tasks where failure is disastrous. Our paper aims to chart this field more concisely by conceptually analyzing the different approaches, a systematic literature review, and benchmarking the difference between approaches on two common control tasks. However, from the reviews (in particular by “9Fsx” and “xEjt”), we understand that we still needed to clarify the motivation for and definition of provably safe RL, as well as, present our experimental results more accurately.

We incorporated all change requests to the best of our knowledge. The main changes for the revised version are:

- We revised the abstract, introduction, section 1.1. and conclusion so that our definition of provably safe RL is clearer and understandable for the reader from the start
- We more explicitly stated the differences between our work and Brunke et al. 2022 and extended the discussion on safety notions in section 1.1.
- We provided more information on the verification methods used for provably safe RL in section 3.
- We improved the section on continuous action masking.
- We added a paragraph to explain our choice of benchmarks and experiments.
- We improved the significance of our experimental results by doubling the number of random seeds and reporting the confidence of the mean.
- We changed the figures reporting experimental results so that it is more obvious how they support our findings.
- We restructured and revised the discussion (section 5) and clarified the limitations and benefits of the proposed taxonomy and the future research directions within this section.

Shortly, we will post answers to the individual reviewer comments and are happy to clarify any follow-up questions or suggestions.

---

### Decision · Action_Editors · 2023-10-16

**Recommendation:** Accept with minor revision

**Comment:**

I am recommending that the paper be accepted with minor revisions. The authors addressed several initial shortcomings pointed by reviewers, and improved the presentation and organization of the paper. For completeness, I suggest that authors cite and contextualize a few more papers from RL + logic and formal verification communities, since provable safety is a core requirement in this survey. Here are a few examples:

LTL and Beyond: Formal Languages for Reward Function Specification in Reinforcement Learning. Camacho et al, IJCAI '19

LTL Realizability via Safety and Reachability Games. Camacho et al, IJCAI '18

A Symbolic Approach to Safety LTL Synthesis. Zhu et al. HVC '17

Policy Synthesis and Reinforcement Learning for Discounted LTL*. Alur et al. CAV '23

Reinforcement Learning With Temporal Logic Rewards. Li et al. IROS '17

Q-Learning for Robust Satisfaction of Signal Temporal Logic Specifications. Aksaray et al. CDC '16

On the (In)Tractability of Reinforcement Learning for LTL Objectives. Yang et al. IJCAI '22

There are multiple other papers that one could cite here, but I'm hoping the authors will manage to present this literature in a more complete way. Other than that, the paper is ready to be accepted. Thank you to the authors for addressing all the issues that the reviewers brought forth.

**Audience:**

The audience for this paper comprises RL researchers and users of RL methods, for whom provable safety guarantees are a hard requirement.

**Claims And Evidence:**

This paper surveys RL methods with provable safety guarantees, namely settings where there is a classifier that verifies whether a state and action tuple is safe with probability 1. Notably, this excludes RL methods that allow mistakes/accidents during exploration in order to learn this classifier from online experience. All methods cited are categorized into action projection, action masking, or action replacement approaches, which is a useful way to organize this literature. The paper also includes benchmarks of popular model-free RL algorithms and how they trade off performance vs safety in the three settings described above. The benchmarks are performed only on two environments: pendulum and 2D quadrotor control, which is admitedly a limited set of tasks, with low-dimensional states. Despite the lack of exhaustive coverage of environments that normally would be expected if the paper were proposing a new RL method, the existing experiments in the paper provide sufficient insight for readers to get some initial empirical evidence into the tradeoffs made by the categorization mentioned above.

---

> ### Author Response · Authors · 2023-11-09
> **Response to Decision**
>
> Dear Editor and Reviewers,
>
> Thanks a lot for your valuable suggestions and feedback. We are happy about the positive decision and incorporated the remaining minor suggestions.
>
> In summary, we made these changes for the camera-ready version:
>
> - We split the “soft constraints” paragraph in Sec. 1.1. into two paragraphs and added the suggested line of literature where temporal logic is used to systematically synthesize the reward.
> - Related to the editor’s previous suggestion that we should discuss the impact of the three provably safe RL classes on exploration strategies, we now gathered the previously scattered discussion on this in a new section in the conceptual analysis part of the paper: Sec. 2.4. This section discusses how the three classes change the probability density function of the action output. We added Figure 2 to visualize this.
> - Based on the editor’s questions, we decided to have a closer look at the “safety of a system” paragraph again. This resulted in improving the definitions and notation, clearly separating the definition from the information that provides a graspable intuition for the definitions, and adapting the assumption to a proposition.
> - We added the link to our reproducible CodeOcean capsule in the paper and improved Sec. 4.1. and Sec. 4.2. by more clearly differentiating when we use the continuous and the discrete system.
> - We corrected minor grammatical errors, spelling mistakes, and improved the notation.
>
> Finally, we want to thank all reviewers and the editor again for this review. We believe our paper significantly benefited from it and we enjoyed the discussions.